# Epstein–Barr virus induces aberrant B cell migration and diapedesis via FAK-dependent chemotaxis pathways

Susanne Delecluse[1,2,3,4], Francesco Baccianti [ID][1,2], Manon Zala[5,6], Alina Steffens[1,2,7], Carolin Drenda[8], Daniel Judt[1,2], Tim Holland-Letz[8], Remy Poirey[1,2], Pierre Sujobert [ID][5,6] & Henri-Jacques Delecluse [ID][1,2] [✉]

Infection with the Epstein-Barr virus (EBV) is a major risk factor for the development of cancer and autoimmune disorders. The virus enters the body in the pharynx, but EBV causes disease in distant organs, including the gut and the brain. Here we show, using in vitro culture and mouse infection models, that EBV-infected B cells display features of homing cells. Infected B cells undergo migration following paracrine CCL4 release and CCR1 induction, while CCR1 deficiency inhibits migration and, unexpectedly, proliferation of infected B cells. Furthermore, migrating EBV-infected B cells undergo CCL4-dependent diapedesis, induce ICAM-1 on endothelial cells, and disrupt the integrity of endothelial barriers. Both migration and diapedesis are regulated by FAK, with FAK inhibition blocking growth and survival of EBV-transformed B cells, as well as their spreading to spleen and brain in an animal model in vivo. Moreover, IL-10 secreted by EBV-infected B cells attracts and facilitates diapedesis of EBV-negative CD52[high]CD11c[+] B cells, which have reported autoimmune properties. Our results thus provide mechanistic insight on EBV-induced B cell dysregulation, and also hint curbing migration as a potential target for reducing the pathogenicity of EBV-infected B cells.

The Epstein–Barr virus was the first discovered human tumor virus[1]. It infects the large majority of the population and persists for life in infected hosts[1]. We currently lack preventative vaccines against this virus and infected cells cannot be eliminated. EBV mainly infects B cells and causes multiple types of lymphoproliferations, e.g posttransplant lymphoproliferative disorders (PTLD), Burkitt's lymphoma or Hodgkin's lymphoma[1].

Primary infection with the Epstein–Barr virus (EBV) can also cause an infectious mononucleosis (IM) syndrome, a self-limiting lymphoproliferation[1]. An EBV infection, particularly if complicated by an infectious mononucleosis, increases the risk of developing multiple sclerosis (MS), the most frequent demyelinating disease, over 30-fold[2]. Moreover, this infection typically precedes by several years the development of neurological lesions in patients with MS. Conversely, individuals who do not carry EBV have a negligible risk of developing MS. Altogether, this points towards EBV infection as a major event in MS pathogenesis. EBV is also suspected to be implicated in the pathogenesis of systemic lupus erythematosus and of other autoimmune diseases[3].

EBV is thought to enter the body in the oropharynx[1]. Accordingly, tonsils are frequently infiltrated by infected B blasts during infectious mononucleosis syndrome or in PTLD that arise in transplanted children and low numbers of infected resting B cells are commonly found

[1]Unit D400, DKFZ, Heidelberg, Germany. [2]Inserm joint unit, Heidelberg, Germany. [3]Department Nephrology, University of Heidelberg, Heidelberg, Germany. [4]German Center for Infection Research (DZIF), Braunschweig, Germany. [5]Centre International de Recherche en Infectiologie (Team LIB), Université Lyon, INSERM, U1111, Université Claude Bernard Lyon 1, Centre National de la Recherche Scientifique, UMR5308, ENS de Lyon, Lyon, France. [6]Faculté de Médecine Lyon-Sud, Université de Lyon, Oullins, France. [7]University of Heidelberg, Heidelberg, Germany. [8]Biostatistics Unit C60, DKFZ, Heidelberg, Germany. [✉]e-mail: h.delecluse@dkfz.de

in tonsils from healthy sero-positive individuals[4–6]. However, EBV-infected B cells can also be detected at distant anatomical sites, for example in lymph nodes, or in the gut-associated lymphoid tissue[7,8]. Furthermore, PTLDs develop frequently in the liver and GI tract of adult transplant recipients and in some transplanted children[5,9]. EBV-infected B cells are found in the brain of healthy individuals, but are more abundant in the brain of individuals with MS[10,11]. These observations imply that infected B cells must be able to migrate outside the lymphoid tissues they initially colonized.

Normal B cells migrate extensively, in particular during development, germinal center maturation and homing to sites of inflammation[12,13]. However, this migration is controlled by multiple chemokines typically secreted by stromal cells or inflammatory cells that interact with cognate receptors at the surface of B cells.

EBV-infected B cells secrete multiple chemokines including CCL3, CLL4, CCL5, CCL22, or CXCL8, but also chemokine receptors such as CCR1 or CCR7[14–16]. Interestingly, CCR1 is a receptor for CCL3 and for CCL4 secreted by lymphocytes[17]. Thus, EBV-infected cells could in principal initiate an autocrine chemotactic or chemokinetic loop. Chemokine receptors are G-protein-coupled proteins that upon ligand binding activate the SRC and FAK kinases to engage the MAP kinase signaling pathway[18].

Here we report that cytokines secreted by B cells upon EBV infection induce migration and diapedesis of both infected and resting B cells. In infected B cells this process is driven by CCL4 that activates CCR1 and FAK in an autocrine manner. Inhibition of this pathway also blocks proliferation of infected B cells, establishing a link between B cell migration and proliferation. Moreover, Il-10 produced by infected B cells recruits resting B cells and allows their diapedesis, in the absence of specific chemotactic signals. Given the importance of endothelial cell barriers to maintain the integrity of most organs, we anticipate that these properties of infected B cells substantially contribute to the pathogenesis of EBV-associated diseases whilst offering new therapeutic opportunities against them.

## Results

### EBV-infected B cells display features of directional migration

We determined the migrating abilities of various types of B cells embedded in a 3D collagen mesh using time lapse microscopy. We first generated an in vitro model of chemokine-induced migration by exposing primary B cells to CD40L, IL-4 and CXCL12 (referred hereafter to as B blasts) to serve as a positive control. Both these blasts and EBV-infected B cells moved with a high average velocity (8 μm/min) (Supplementary Movie 1 and 2, Fig. 1a–c). In contrast, resting B cells and B cells stimulated with CD40L and IL-4 alone were nearly immobile (Supplementary Movie 3 and 4, Fig. 1a–c). The tracks formed by the migrating infected B cells and migrating B blasts consisted of long linear segments separated by a few angles, suggesting that they were predominantly directed, i.e. ballistic in nature, and not purely random Brownian motions[19,20]. The recorded trajectories of these single cells displayed an intermediate degree of directionality for both B cell subtypes (60%), supporting this hypothesis (Fig. 1d, see Supplementary Note 1 for a detailed explanation). We then generated mean displacement plots from for the B cell subtypes (Fig. 1e, see Supplementary Note 1 for a detailed explanation)[20]. While the movement of control B cells generated a linear graph that is characteristic of a random Brownian migration, EBV-infected cells and blasts covered larger distances per time unit and generated a 'sublinear' graph that is again indicative of a sustained directed migration (Fig. 1e). We next developed a correlated random walk simulation program that generates walks with various degrees of directionality (from purely random to purely directed). This model confirmed that EBV-infected cells and B blasts have predominantly directed paths (60 and 70%, respectively) that significantly deviate from purely random walks ($p < 0.001$) (Fig. 1f)[20].

### Migration of EBV-infected cells depends on their concentration and is governed by CCL4 and CCR1

Directed paths are generated by lymphoid cells under chemotactic or chemokinetic stimulation[19]. However, because the paths generated by EBV-infected B cells do not converge, these cells are subjected to chemokinesis rather than chemotaxis. This suggests that the concentration of chemokines in the extracellular milieu is homogeneous, rather than concentrated in a discrete region of the culture. Thus, the chemotactic signals in a culture of EBV-infected B cells probably originate from many members of the cell culture, if not all of them. These observations prompted us to canvass the literature for cytokines or chemokines that are expressed together with their receptor by EBV-infected cells. We identified human IL-10 (hIL-10), EBV-encoded IL-10 homolog (also referred to as viral IL-10 or vIL-10), CCL3, CCL4 and CCL5 as potential candidates[21]. ELISA-based assays confirmed that all these cytokines are secreted in the supernatant of EBV-infected B cells, with CCL3 and CCL4 being secreted at concentrations more than hundred times higher than in supernatants from primary B cell controls (Supplementary Fig. 1a). Chemotaxis chamber assays with these cytokines showed that they all attract infected B cells, with CCL4 showing the strongest effects (Fig. 2a, Supplementary Fig. 1b). Moreover, when EBV-infected B cells embedded in a collagen mesh were subjected to a CCL4 gradient, they clearly moved towards increasing concentrations of the chemokine (Supplementary Fig. 1c). Reciprocally, incubation of infected B cells with neutralizing antibodies specific to CCL4 approximately halved the proportion of infected B cells undergoing migration (Supplementary Fig. 1d, e). Similarly, CXCL12's effect on B blasts' migration largely disappeared after exposure to a CXCR4 inhibitor (AMD3100), confirming that their directional movement resulted from a chemokinetic stimulus (Supplementary Fig. 1f). To clarify CCL4's role in EBV-infected B cells, we knocked out its gene by CRISPR/Cas9. The CCL4[null] cells lost 90% of their ability to migrate and did not grow, except if they were supplemented with exogenous CCL4 (Fig. 2b, Supplementary Fig. 2a). If cell movement is dependent on chemokine secretion and the infected cells themselves produce the chemokines, infected cells cultured at low concentration should move less efficiently. Therefore, we repeated the above-described experiments with EBV-infected B cells seeded at low cell density ($3 \times 10E4$/ml). Under these conditions, the majority of infected B cells were immobile and the few cells in motion showed short and slow trajectories (Fig. 2c, d and Supplementary Movie 5). Thus, infected cells could move only if they were close enough to other infected B cells. However, if exogenous CCL4 was added to the culture medium, cells resumed migration and became indistinguishable from cells seeded at higher concentration (Fig. 2c, d). Altogether, CCL4 appears to play a central role in migration of EBV-infected cells. We then attempted to inhibit CCR1, the only CCL4 receptor expressed by infected cells (Supplementary Fig. 3a, b)[16]. Treatment of infected cells with the CCR1 antagonist BX471 efficiently reduced the percentage of migrating cells and their speed, but only at concentrations above 10 μM (Supplementary Fig. 2b and Supplementary Fig. 3c). This treatment did not increase apoptosis after 24 h, but blocked cell growth and eventually induced cell death (Supplementary Fig. 2b, c). In the same vein, CCR1[null] cells were unable to migrate and grow and progressively died after three weeks in culture (Fig. 2b, Supplementary Fig. 2a). In summary, interaction between CCL4 and CCR1 orchestrates EBV-induced cell migration, growth and survival.

### EBNA2 and LMP1 control CCR1 and CCL4 expression

Latently infected B cells proliferate under expression of the latent genes, among which the Epstein–Barr nuclear antigen 2, a transactivator that activates the Notch pathway and the latent membrane protein 1, a permanently active viral homolog of CD40, play a crucial role[1]. To establish whether these viral genes control CCL4 and CCR1 expression in infected B cells, we first monitored their expression in an

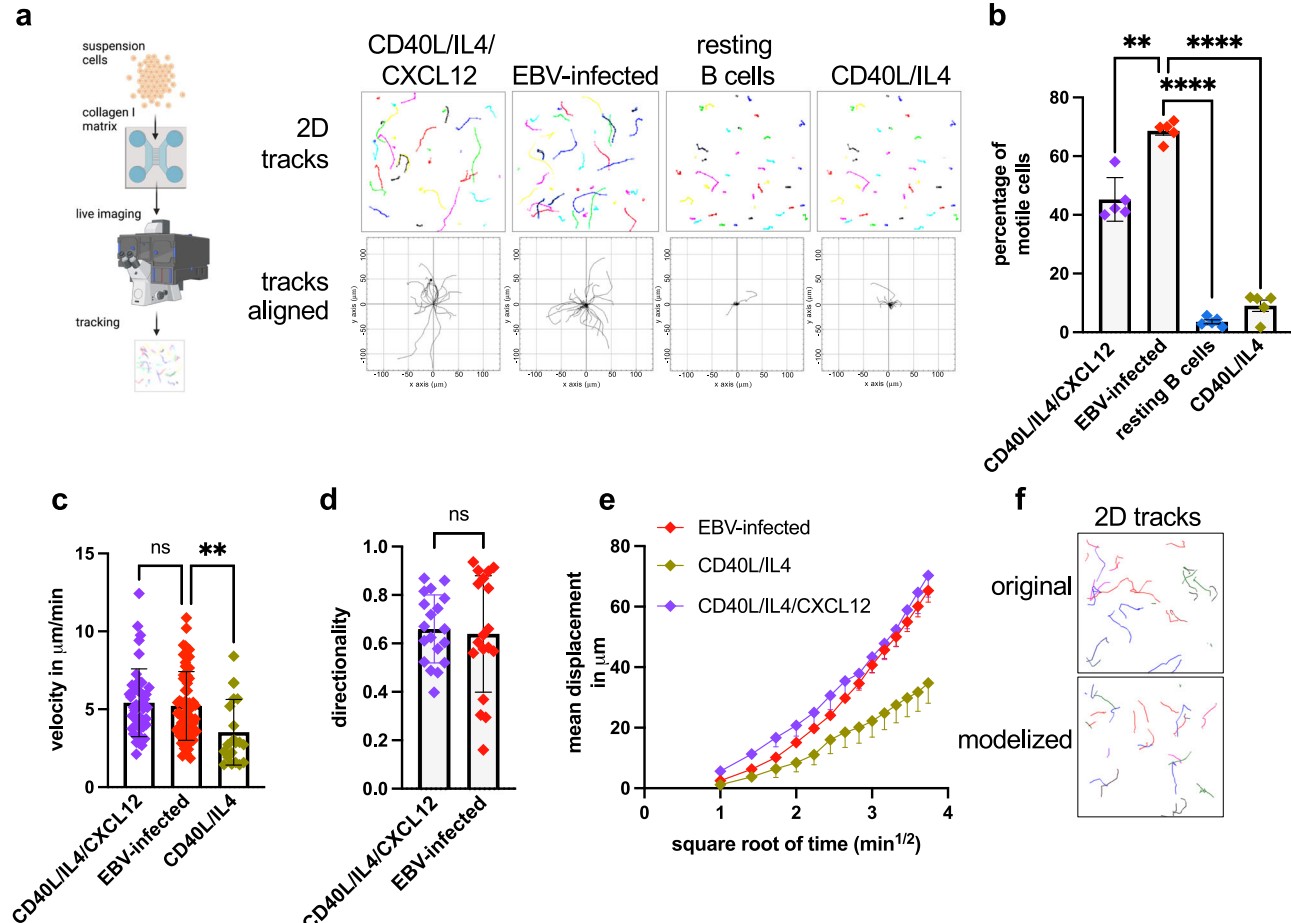

**Fig. 1 | EBV-infected B cells display non-random migration. a–e** B cells stimulated with CD40L, IL-4 and CXCL12, EBV-infected B cells, primary resting B cells or B cells stimulated with CD40L and IL-4 were seeded in a collagen matrix at a concentration of 3 × 10E5 cells/ml and subjected to time lapse microscopy for 15 min ($n = 50$, one representative example from 5 experiments with independent B cell samples). **a** Generated paths were projected onto a 2D surface to generate tracks with or without alignment ($n = 50$, one representative example from 5 experiments with independent B cell samples). Created in BioRender. Poirey, R. (2025) https://BioRender.com/hgal8k4. **b** Analysis of the generated tracks revealed the percentage of motile cells. Bar graphs give the mean and standard deviation ($n = 50$, one representative example from 5 experiments with independent B cell samples). **c** The velocity of migrating cells is given in the bar graph as mean with standard deviation ($n = 50$, one representative example from 5 experiments with independent B cell samples). **d** The directionality of migrating cells is given in the bar graph indicating the mean and standard deviation ($n = 20$, one representative example from 5 experiments with independent B cell samples). **e** Square root time profile of migrating B cells. The graph shows the mean displacement of the cells as a function of the square root of time with mean and standard deviation ($n = 50$, one representative example from 5 experiments with independent B cell samples). **f** The original tracks were compared to a modelized random walk to determine their percentage of directionality. We show 2D tracks generated by EBV-infected B cells, together with modelized paths harboring 70% directionality. Statistical significance was determined using one-way analysis of variance in (**b**) and (**c**) and unpaired two-sided *t* tests in (**d**). **$p < 0.01$, ****$p < 0.0001$. Source data are provided as a Source Data file.

early and a late passage of the MUTU Burkitt's lymphoma cell line. While an early passage of this line (MUTU I) has a restricted latent protein expression pattern largely limited to EBNA1 (latency I), late passage cells (MUTU III) express all latent genes, including EBNA2 and LMP1 (latency III)[22]. CCR1 surface expression and CCL4 release in MUTU III was respectively 8 (CCR1) and 600 (CCL4) times higher than in MUTU I, suggesting that expression of these proteins is associated with latency III (Fig. 2e). To determine which latent gene is responsible for these effects, we transfected EBNA2 or LMP1 in CD40L + IL4 activated B blasts. These assays revealed that LMP1 is a potent inducer of CCL4 and CCR1 expression (Fig. 2f). Interestingly, EBNA2 also activated CCL4 release, but was unable to induce CCR1 expression (Fig. 2f). Similar results were obtained after transfection of the cell line BL41 with LMP1 or EBNA2 (Supplementary Fig. 3d). To confirm LMP1's role in this process, we infected primary B cells with a LMP1null mutant (M81/ΔLMP1). Relative to B cells infected with wild-type viruses, B cells infected with M81/ΔLMP1 released CCL4 and expressed CCR1 at 10- and 5-time lower levels, respectively (Fig. 2g). This approach was unfortunately not possible for EBNA2 as the M81/ΔEBNA2 mutant fails

to initiate transformation. Altogether, we conclude that LMP1 and EBNA2 collaborate to initiate the chemokine loop that leads to migration.

## EBV-infected B cells display morphological features of migrating cells

Examination of our B cell panel by scanning electron microscopy (SEM) or confocal light microscopy combined to a membrane dye revealed that both EBV-infected B cells and our control B blasts displayed typical features of migrating B cells (Fig. 2h, Supplementary Fig. 4a). These cells displayed a well-developed uropode at one pole, and a large lammelipodia intertwined with filopodiae at the opposite pole. These features were visible in the control blasts, but not in the negative controls (Supplementary Fig. 4a). Treatment of EBV-infected B cells with specific inhibitors showed that polarization results from F-actin polymerization and requires myosin II (Supplementary Fig. 5a, b). Lamellipodiae drove migration of both infected B cells and control B blasts (Supplementary Movie 1 and 2). We found that CCR1 is preferentially expressed at the surface of the lamellipodiae

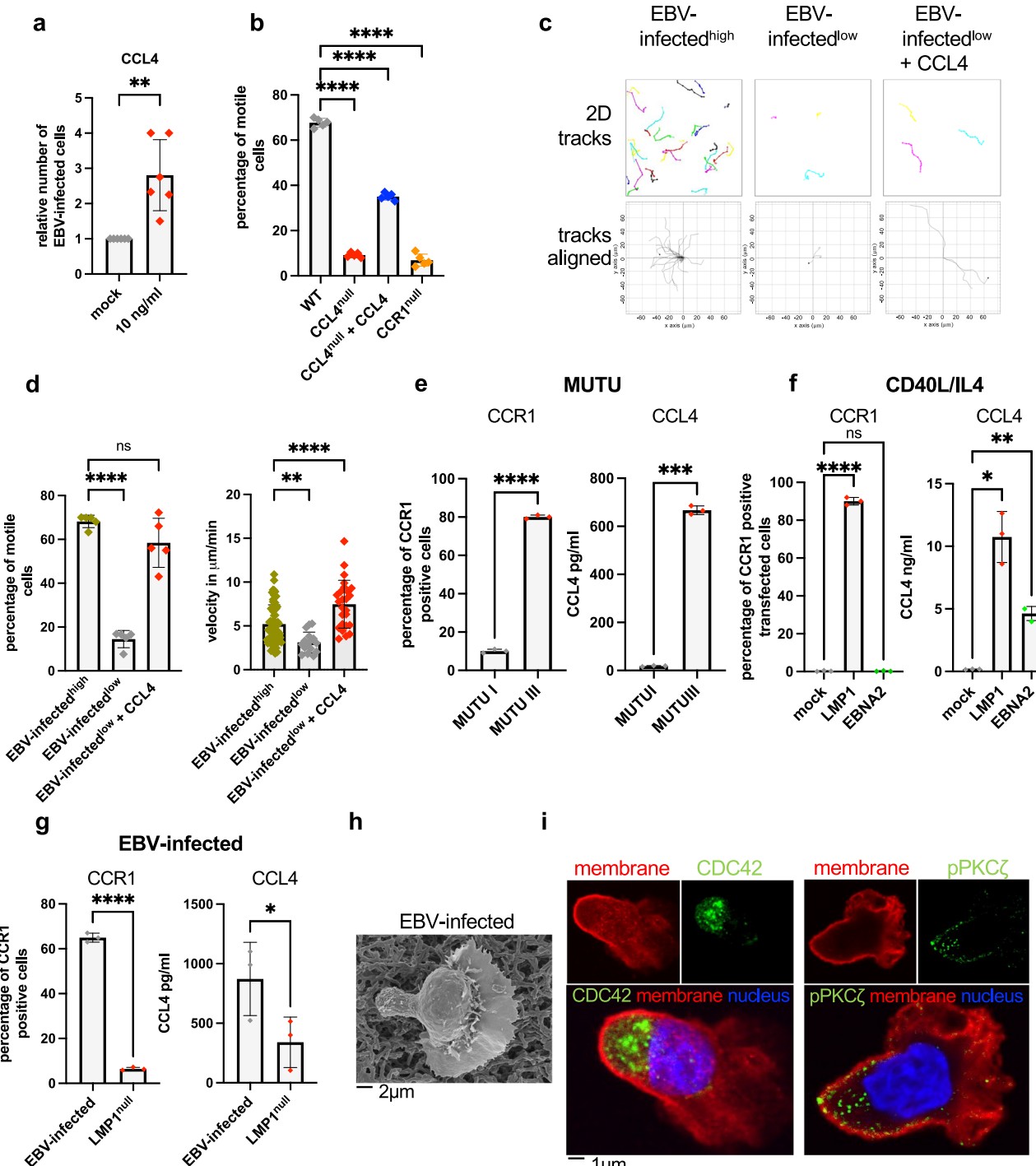

(Supplementary Fig. 3b). Morphological features of polarization were confirmed by the study of specific molecular markers. Immunofluorescence staining showed that the small G protein CDC42 or phosphoPKC Zeta were concentrated between the uropode and the nucleus of infected B cells, but were randomly distributed in B cell controls (Fig. 2i and Supplementary Fig. 4b). This is relevant as these molecules induce polarization and are involved in cell migration[23,24]. Furthermore, centrin and live tubulin stains showed that the microtubule organizing center (MTOC) was also located in this area, as were cellular organelles such as mitochondria or Golgi apparatus (Supplementary Fig. 4c–e)[23]. Similar features were observed in our control B blasts, but not in non-moving controls (Supplementary Fig. 4b–e). Interestingly, polarization and lamellipodia developed slowly post-infection and reached their full development only 2 weeks after

infection, a time at which cells start to migrate and proliferate efficiently (Supplementary Fig. 4f). After that time point, these morphological features and the ability to migrate persisted unchanged, even after several months in culture.

## Migration of EBV-infected B cells is governed by the FAK2-CDC42-ROCK pathway

G protein-coupled receptors such as CCR1 have been reported to signal through the FAK pathway with which CDC42 and the Rho-associated kinase (ROCK) interact[18,25,26]. EBV-infected cells treated with FAK2, CDC42 or ROCK inhibitors lost polarization and evinced markedly reduced mobility (Supplementary Movie 6, Fig. 3a, Supplementary Fig. 5a, b). The FAK2 inhibitor defactinib was particularly interesting as it immediately and strongly reduced cell mobility and

**Fig. 2 | Migration of EBV-infected B cells is driven by CCL4 and CCR1. a** Transwell assay with CCL4 as an attractor. $5 \times 10E4$ infected B cells were seeded in the top chamber. Medium containing CCL4 (10 ng/ml) was placed in the bottom chamber. The number of infected cells that have reached the bottom chamber after 1 h of stimulation is given, relative to the number of spontaneously migrating cells ($2.5 \times 10E3$). Medium devoid of chemokines served as a negative control. The bar graph gives the mean with standard deviation ($n = 6$ independent B cell samples). **b** The bar graph shows the mean percentage and standard deviation of motile CCL4[null] or CCR1[null] EBV-infected cells as observed by time-lapse microscopy ($n = 5$ independent B cell samples). The impact of exogenous CCL4 on the mobility of CCL4[null] cells is also shown. **c** EBV-infected B cells were seeded in a collagen matrix at low concentration ($3 \times 10E4$ cells/ml) in the presence or absence of CCL4. Cells were subjected to live cell imaging for 15 min to generate 2D tracks with or without alignment ($n = 50$ one out of 5 individual experiments). The same cells seeded at high concentration ($3 \times 10E5$ cells/ml) served as a positive control. **d** We determined the percentage and the velocity of mobile cells analyzed in (**c**). Results are given as bar graphs with mean and standard deviation ($n = 50$ one out of 5 individual experiments). **e–g** EBV latent genes and CCL4-CCR1 expression. **e** CCR1 surface expression on Burkitt cell line MUTU I and III clones was determined by FACS

(left panel) and CCL4 concentration in supernatants from these cells was determined by ELISA (right panel) ($n = 3$ independent transfections). The results are summarized in bar graphs (mean with standard deviation). **f** B cells stimulated with CD40L + IL-4 were transfected to express LMP1 or EBNA2. The bar graphs show CCR1 surface expression and CCL4 release in transfected cells and in empty vector controls. Results were normalized for the percentage of transfected cells. Bar graphs show mean and standard error of the mean ($n = 3$ transfected independent B cell samples). **g** Differential CCR1 expression at the surface of B cells infected with a LMP1[null] virus or wild type controls (left panel) and CCL4 release in the supernatants of these cells (right panel) ($n = 3$ independent B cell samples). The results are summarized in bar graphs (mean with standard deviation). **h** EBV-infected B cells were subjected to scanning electron microscopy. **i** EBV-infected B cells were pre-fixed in PFA and stained with antibodies specific to CDC42 and phosphoPKC Zeta (green), together with a membrane dye (red) and nuclear dye DAPI (blue). Representative pictures are shown ($n = 30$, one representative example from 3 experiments with independent B cell samples). Statistical significance was determined using two-sided paired $t$ tests in (**a**), (**e**), and (**g**) and one-way analysis of variance in (**b**), (**d**), and (**f**). $*p < 0.05$, $**p < 0.01$, $***p < 0.001$, and $****p < 0.0001$. Source data are provided as a Source Data file.

cell growth, but also cell survival after 2 weeks at moderate concentrations (3.5 μM) (Fig. 3a). Defactinib has previously been tested in clinical trials at doses delivering similar seric concentrations[27,28]. Defactinib induced low levels of apoptosis after one day at this concentration (Supplementary Fig. 2c). Defactinib did not affect growth of activated non-infected B cells (Supplementary Fig. 2d). Interestingly, EBV-infected B cells expressed phosphoFAK2 at levels three times higher than those of the stimulated B blasts (Fig. 3b). This suggests that growth of EBV-infected cells is dependent on higher phosphoFAK2 expression levels and thus is more sensitive to defactinib (Fig. 3b). Because defactinib inhibits both FAK1 and FAK2, we assessed expression of these proteins in EBV-infected B cells. Immunostains revealed that these cells nearly exclusively express the FAK2 protein, although some residual expression of the non-lymphoid kinase FAK1 was visible (Supplementary Fig. 6). Moreover, while FAK2 was localized at the uropode, FAK1 was mainly perinuclear in location. We then assessed the effect of CCR1 inhibition on phosphoFAK2 expression levels in infected B cells. Exposure of infected B cells to CCR1 inhibitors at 10 μM more than halved phosphoFAK2 expression levels (Fig. 3c). Although low doses of the CCR1 inhibitor BX471 (5 μM) and of defactinib (0.5 μM) were unable to influence growth of infected B cells, their combination showed clear synergistic effects (Fig. 3d, Supplementary Fig. 2e). We confirmed synergy between BX471 and defactinib first by drawing a dose-response curve for these drugs (Fig. 3e). This analysis showed a very steep dose-response slope for the latter drug that limited the range of concentrations that could be used to draw isobolograms. Nevertheless, isobolograms for $E_{10}$ (10% of complete growth reduction) and $E_{30}$ (30% of complete growth reduction) showed clear synergistic effects (Fig. 3e). At higher doses, both drugs were individually able to reduce cell growth. Under these conditions, as expected, synergistic effects could not be observed. Altogether, these data confirm that CCR1 and FAK2 are located in the same pathway and are essential for EBV-infected B cell polarization and cell migration.

## EBV-infected B cells attract CD52high-CD11c+ resting B cells

Autoimmune diseases associated with EBV, and in particular MS, frequently show tissular infiltration with immune cells, but the cause of their recruitment is not clear[29]. Thus, we investigated whether the large amounts of chemokines secreted by EBV-infected B cells could influence movement of non-infected lymphoid cells. Migration assays indeed showed that EBV-infected B cells attract primary B cells, but not activated B cells (Fig. 4a, Supplementary Fig. 7a). Next, we attempted to identify the cytokines responsible for this effect. Because primary B cells express the IL-10 receptor but not the receptors for CCL3, CCL4 and CCL5, we performed chemotaxis assays with human IL-10 and EBV-encoded IL-10. These cytokines increased migration of resting, but not

activated control B cells (Fig. 4b, Supplementary Fig. 7a). Reciprocally, an EBV-infected B cell line that lacks both EBV-encoded and human IL-10 (Fig. 4b) lost its ability to attract resting B cells. Although IL-10 does not act as a chemotactic factor, it has been reported to modulate the action of chemokines and to increase chemokinesis in B cells[30]. Indeed, incubation of primary B cells with a mix of EBV-encoded and human IL-10 or coculture with EBV-transformed cells led to an increased proportion of migrating cells (Supplementary Fig. 7b). In an attempt to characterize the mobile B cell population attracted by EBV-infected B cells, we performed single-cell RNAseq (scRNAseq) and FACS stains on the mobile (move) and immobile (stay) subpopulations. UMAP visualization based on clustering analysis identified 3 clusters within the primary B cell population. These were defined on the basis of a NF-kB activation signature, of a BCR response with SYK expression or of a combined IL-4R and IL-7R expression, but we did not find any evidence that this clustering differed between the move and stay population (Fig. 4c, d). However, gene set enrichment analysis (GSEA) revealed that moving primary B cells strongly express interferon alpha and gamma response genes, which probably results from the contact with infected B cells that produce interferons (Fig. 4e)[31,32], as well as an increased oxidative phosphorylation and a decreased entry into mitosis. ScRNA analysis also identified CD52 and CD53 expression as enhanced in the majority of migrating B cells, relative to their immobile counterparts (Fig. 4f, Supplementary Table 2). We could confirm that CD52, a protein involved in endothelial diapedesis, is more strongly expressed in the migrating cells, although the difference in expression between the migrating and non-migrating populations was limited in intensity (Fig. 4f)[33]. Altogether, this suggests that the motile B cell subset is metabolically active and primed for migration and diapedesis. Furthermore, we found that migrating cells were enriched in CD11c-positive cells, relative to the initial B cell population (Fig. 4g). However, coculture of primary B cells with infected B cells could not reproduce this effect, excluding that contact with EBV-infected B cells upregulated CD11c (Supplementary Fig. 7c). CD11c is a marker of atypical B cells that play an important role in autoimmune diseases and are enriched in the brain of patients with MS[34,35].

## EBV-infected B cells breach endothelial barriers and allow B-cell diapedesis

Lymphoid cells homing to tissues under chemotactic stimulus perform diapedesis to cross endothelial barriers and reach, for example, inflamed tissues[36]. We first used a transwell assay in which the upper and lower chambers are separated by a layer of activated human endothelial cells derived from brain, dermis, or umbilical vein (HBMEC, HDMEC or HUVEC) cells. HBMEC cells are typically used to model the blood-brain barrier (BBB)[37]. Addition of infected B cells to the top

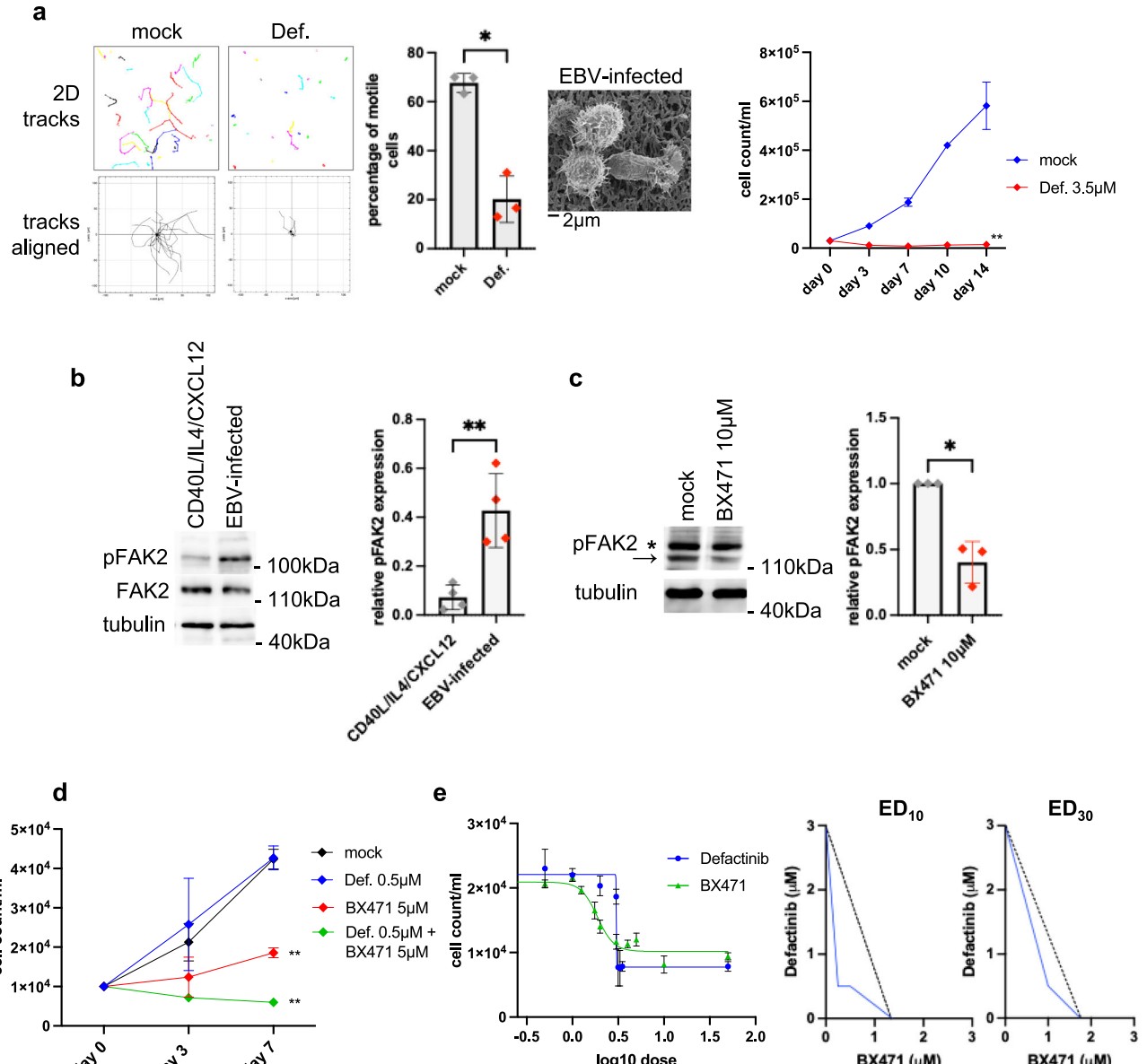

**Fig. 3 | Migration of EBV-infected B cells is dependent on FAK2. a** EBV-infected B cells were exposed to the FAK2 inhibitor defactinib (Def.) for 30 min (3.5 μM) and seeded in collagen at a concentration of 3 × 10E5 cells/ml. 2D cell tracks with or without alignment were generated from time lapse microscopy recordings. The bar graphs show the mean percentage of motile cells with standard deviation in treated and untreated cells (n = 3 independent B cell samples). Treated EBV-infected B cells were also observed by scanning electron microscopy. Curves showing EBV-infected cell growth in the presence of defactinib (Def.) at a 3.5 μM concentration over two weeks. Mean cell count and standard deviation are indicated (n = 5 independent B cell samples). **b** Total and phosphoFAK2 expression levels in B cells stimulated with CD40L, IL-4 and CXCL12 or after EBV infection were determined by western blot with specific antibodies. The bar graph summarizes the mean phosphoFAK2 expression levels and standard deviation, relative to the loading control tubulin (n = 4 independent B cell samples). **c** Western blot showing phosphoFAK2 expression levels in EBV-infected B cells exposed to the CCR1 inhibitor BX471 at 10 μM for 30 min. The arrow refers to phosphoFAK2, the asterisk refers to a non-specific signal. The bar graph summarizes the mean phosphoFAK2 expression levels and standard deviation. Values are given relative to the values observed in the untreated population. An immunoblot against tubulin was used as loading control (n = 4 independent B cell samples). **d** Curves showing EBV-infected cell growth over time in the presence of defactinib (Def.) at a 0.5 μM concentration. Infected cells were alternatively treated with BX471 (5 μM) alone or with a combination of low doses defactinib (0.5 μM) and BX471 (5 μM). Mean cell count and standard deviation are indicated (n = 5 independent B cell samples). **e** dose-response curve for defactinib and BX471 (n = 3 independent B cell samples, each point showing mean and standard deviation), together with isobolograms (ED$_{10}$ = 10% of maximal effect and ED$_{30}$ = 30% of maximal effect). The linear curve shows additivity of the drug effects, the curve below it is indicative of synergistic effects. Statistical significance was determined using two-sided paired *t* tests in (**a**) (growth curve at day 14), **b** and (**c**) and one-way analysis of variance at day 7 in (**d**). *$p < 0.05$, **$p < 0.01$. Source data are provided as a Source Data file.

chamber led to an efficient crossing of the endothelial barrier within 24 h with all three types of endothelia (Fig. 5a, Supplementary Fig. 7d). Diapedesis of primary B cells, CD40L + IL-4-stimulated B cells and CXCL12-activated B blasts through this HUVEC or HBMEC barrier was ten-fold less efficient (Fig. 5a, Supplementary Fig. 7d). Diapedesis of EBV-infected B cells through HUVEC and HBMEC cells increased

endothelial permeability, as assessed by a decreased trans-endothelial electrical resistance (TEER) (Supplementary Fig. 8a, b). We then tested the ability of EBV-infected B cells to adhere to endothelial layers stimulated with TNF-α under constant laminar shear stress. EBV-infected B cells, and to a lesser extent CXCL12-stimulated B blasts, adhered strongly to both types of endothelia and showed evidence of rolling in

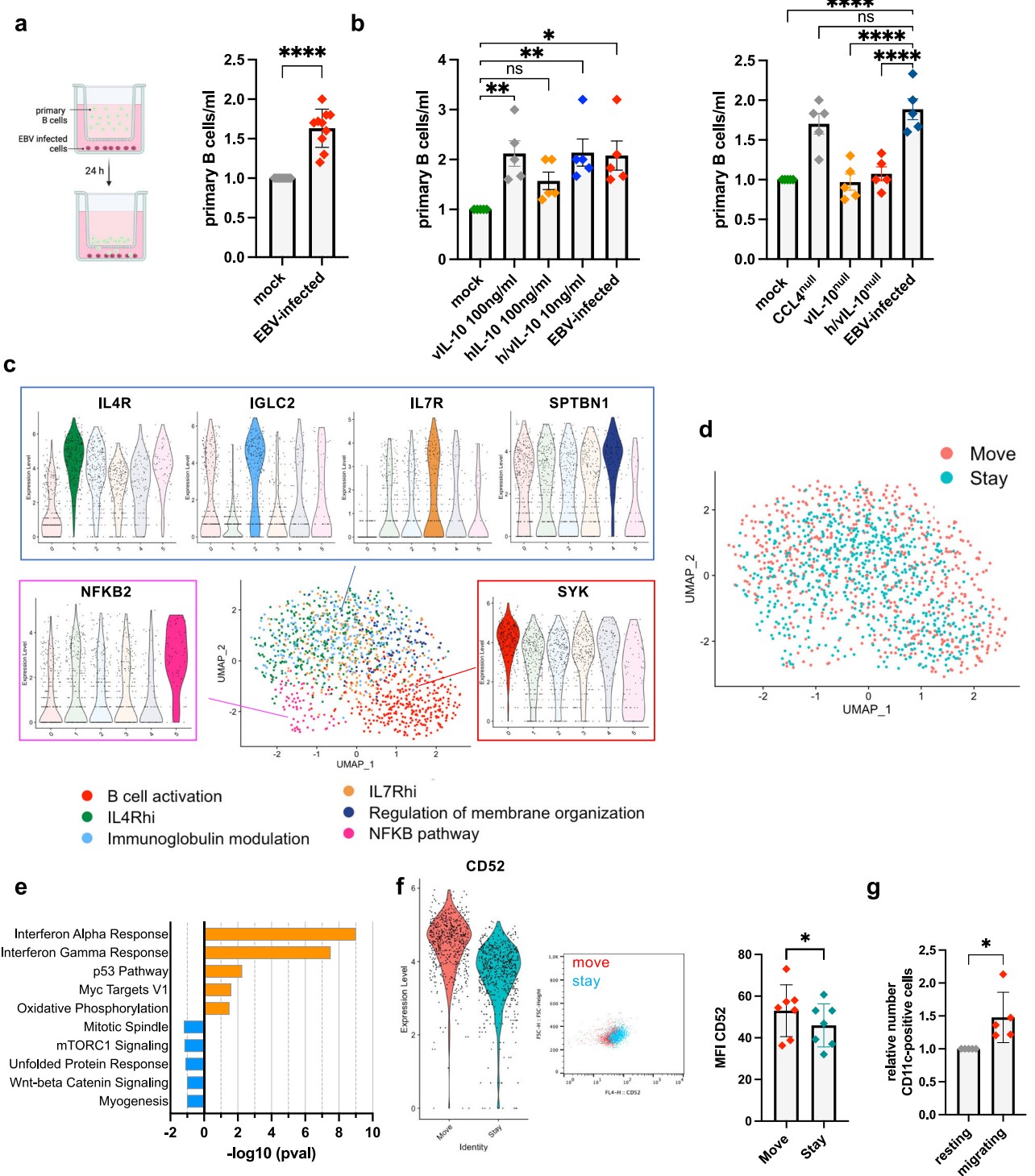

these flow experiments (Fig. 5b, Supplementary Movie 7, Supplementary Fig. 8c, d). Moreover, only EBV-infected B cells bound to unstimulated HBMEC, albeit at very reduced levels (Fig. 5b). However, EBV-infected B cells treated with defactinib lost their ability to adhere to the endothelium (Fig. 5b, Supplementary Fig. 8c). Coculture of confluent endothelial cells with infected B cells, but not with CXCL12-activated B blasts, activated calcium signaling within 10 min, as expected from endothelia experiencing diapedesis (Supplementary Fig. 9a)[38]. Coculture of EBV-transformed B cells with the three types of endothelial barriers led to a relocation of the zona occludens protein 1 (ZO-1), a marker of endothelial barrier integrity, from the intercellular areas to the cytoplasm, and to a disruption of the cytoskeletal structure around

cell-to-cell contact areas (Fig. 5c, Supplementary Fig. 9b)[39]. Rolling of EBV-infected B cells on endothelial cells was followed by a loosening of endothelial junctions within 10 min, with the development of large clefts, as shown by live labeling of the F-actin network (Supplementary Fig. 9c, d). This suggested that EBV-infected B cells perform transcytosis.

### Chemokines produced by EBV-infected B cells upregulate ICAM-1 expression to induce diapedesis

Diapedesis requires interactions between surface molecules, e.g. the integrin LPAM-1 expressed by lymphoid cells that home to the GALT and MAdCAM-1, an addressin expressed by high endothelial cells

**Fig. 4 | EBV-infected B cells attract CD52$^{high}$ CD11c$^+$primary B cells. a** Transwell migration assay with EBV-infected B cells in the bottom well and labeled primary resting B cells in the top well. The graph shows the mean number and the standard deviation of labeled primary B cells that reached the bottom chamber after 24 h, relative to spontaneous B cell migration in the absence of infected B cells that served as a negative control ($n = 5$ independent B cell samples, each experiment in duplicate). On average approximately $2.5 \times 10$ E3 primary B cells spontaneously migrated to the bottom chamber. Created in BioRender. Poirey, R. (2025) https://BioRender.com/qnmhdf1. **b** Primary B cells were placed in the top chamber of a transwell device and human IL-10 (hIL-10, 100 ng/ml) or EBV-encoded IL-10 (vIL-10, 100 ng/ml) or both cytokines combined (10 ng/ml each) were added to the bottom chamber. EBV-infected B cells placed in the bottom chamber served as a positive control. The number of primary B cells that reached the bottom chamber after 24 h is indicated relative to spontaneous B cell migration (left panel, $n = 7$). Same experiment as in (**a**) was repeated with B cells transformed with a virus that lacks CCL4 (CCL4$^{null}$), EBV-encoded IL-10 (vIL-10$^{null}$) or viral and human IL-10 (h/vIL-10$^{null}$) placed in the bottom chamber. B cells infected with wild-type EBV served as a positive control (right panel, $n = 5$ independent B cell samples). Bar graphs give the mean and standard deviation. **c** Uniform Manifold

and Projection (UMAP) plot of scRNA-seq analysis performed on all primary B cells and colored by annotation. **d** UMAP visualization of scRNA-seq data from mobile (move) and immobile (stay) B cell populations. **e** Top five enriched (orange) and depleted (blue) Gene Ontology terms in the mobile B cell population, relative to the immobile B cells. **f** The graph shows CD52 differential expression level in the move and stay B cell populations after scRNA analysis. Differential CD52 expression in both cell populations was confirmed by flow cytometry ($n = 7$ independent B cell samples, 2000 cells recorded). The bar graph gives the mean fluorescence intensity (MFI) and standard deviation of CD11c expression in these subpopulations. **g** Same experiment as in (**a**), primary B cells that remained in the top chamber and those that migrated to the bottom chamber were stained for CD11c. The bar graph gives the mean and standard deviation of CD11c expression in these subpopulations, relative to the proportion of CD11c in the initial primary B cell population ($n = 5$ independent B cell samples, 2000 cells recorded). Statistical significance was determined using two-sided paired $t$ tests in (**a**), (**f**) and (**g**), one-way analysis of variance in (**b**), non-parametric two-sided Wilcoxon Rank Sum test in (**e**). $P$-values were adjusted using the Bonferroni correction. $^*p < 0.05$, $^{**}p < 0.01$, $^{****}p < 0.0001$. Source data are provided as a Source Data file.

within this lymphoid tissue[40,41]. In tissues infected by pathogens, chemokines and inflammatory mediators such as TNF-α can also upregulate ICAM-1 and VCAM-1 expression at the surface of endothelial cells[42]. This facilitates binding and diapedesis of activated homing lymphocytes[43]. Exposure of primary endothelial layers to EBV-infected B cells led to a clear increase in ICAM-1, but not VCAM-1 or MAdCAM-1 expression (Fig. 5d, Supplementary Fig. 10a, b). EBV-infected B cells express LFA-1, a protein that efficiently interacts with ICAM-1, suggesting that this interaction might contribute to diapedesis (Supplementary Fig. 10c). We tested this hypothesis by blocking ICAM-1-LFA-1 interactions using the BIRT377 LFA-1 inhibitor (Fig. 5b). Addition of this inhibitor to endothelial cells co-cultured with EBV-infected B cells indeed reduced adherence. We then treated HUVEC with some chemokines produced by EBV-infected B cells. CCL4, human IL-10 and EBV-encoded IL-10 all upregulated ICAM-1 expression, with the latter two inducing ICAM-1 nearly as efficiently as TNF-α, the key cytokine in this process (Supplementary Fig. 10d, e, g). HBMEC spontaneously express low levels of ICAM-1 that increased after CCL4 exposure, but were not sensitive to stimulation with IL-10 (Supplementary Fig. 10d, f, g). Coculture of HUVEC cells with CCL4$^{null}$ or IL-10$^{null}$ infected B cells approximately halved ICAM-1 expression, relative to levels obtained with wild-type cells (Supplementary Fig. 10e, g). Similar results were obtained with HBMEC cells cocultured with CCL4$^{null}$ infected B cells (Supplementary Fig. 10f, g). ICAM's induction by CCL4 is congruent with the ability of this chemokine to disrupt the neurovascular endothelium[44]. CCL4$^{null}$ and double human and viral knockout h/vIL-10$^{null}$ EBV-infected B cells had a reduced propensity to cross a continuous HUVEC layer, confirming their role in diapedesis (Supplementary Fig. 10h). However, this observation could not be reproduced with HBMEC, probably because they spontaneously express low ICAM-1 levels. The observation that EBV-infected B cells attract primary B cells suggests that they might also be able to facilitate their diapedesis. Therefore, we exposed an HBMEC endothelial barrier to EBV-infected B cells admixed with primary B cells. The presence of EBV-infected B cells doubled diapedesis of primary B cells (Fig. 5e). Among the primary B cell population that crossed the endothelial barrier together with infected B cells, CD11c-positive B cells were enriched, relative to the total primary B cells (Fig. 5f).

### A FAK-2 inhibitor blocks B-cell lymphoproliferative disease in a murine model

Because defactinib blocks EBV-infected B cells migration and cell viability in vitro, we tested whether this molecule can prevent EBV-induced B cell migration and growth in immunosuppressed NSG mice. Half of the mice received defactinib two weeks after injection of primary B cells exposed to a low virus dose ($4 \times 10^4$ infected cells per mouse) for a

duration of four weeks (Fig. 6a). This prevents the development of large EBV-induced tumors within the observation period. Six weeks after injection of the infected B cells, all mice were alive, but splenic infiltration by EBV-infected lymphoid cells was obvious only in the untreated population. Mice treated with defactinib did not show evidence of splenic invasion, neither after in situ hybridization on splenic tissue using the EBV-specific non-coding RNA probe EBER, nor after qPCR with EBV-specific probes (Fig. 6b). In the same vein, blood samples drawn from defactinib-treated mice at termination of the experiment were devoid of detectable viral sequences. However, the blood of three out of 5 non-treated infected mice tested positive for the EBV-specific qPCR (Fig. 6b). We also subjected brain and meningeal samples from the two mice groups to the same assay. We found that only mice treated with defactinib were free of detectable virus infection (Fig. 6b). This suggests that EBV-infected B cells can independently reach the central nervous system, though at very low levels.

## Discussion

Understanding the contribution of EBV to the development of autoimmune diseases, and in particular of multiple sclerosis, is seen as key for the understanding of their pathogenesis and the development of targeted therapies[45]. In particular, whether and how EBV accesses the brain and other organs is a central question. In the present paper, we found that EBV infection induces polarization, and confers to these cells the ability to migrate and undergo diapedesis with high efficiency, two cardinal features of homing cells. All these features can be largely explained by expression of the well-characterized CCR1 and CCL4 molecules that induced an unexpected paracrine chemokinesis that allowed cells to move independently from the external stimuli that are required for cell homing in general. This is possible because infected B cells simultaneously induced the expression of a chemokine (CCL4) and of its receptor (CCR1). It is also remarkable that the viral infection activated expression of two molecules that are typically expressed by T cells, although CCR1 has been detected in a subpopulation of non-germinal center B cells and CCL4 is released after B cell receptor stimulation[46,47]. CCR1 binding led to an engagement of the FAK2 pathway, but it remains possible that other stimuli also play a role in its activation in EBV-infected B cells. Importantly, cell migration in the context of EBV infection influenced cell growth and ultimately cell survival. Migration blockers eventually eliminated infected B cells, including those undergoing spontaneous lytic replication, a virus infection mode previously associated with multiple sclerosis[48]. This reflects the effects of G protein-coupled receptors such as CCR1 and its downstream effector FAK2 on cell growth and division and offers new therapeutic perspectives[49–51]. Our data expand the current view on EBV-mediated B cell transformation, in which EBV latent proteins directly interact with signaling pathways to

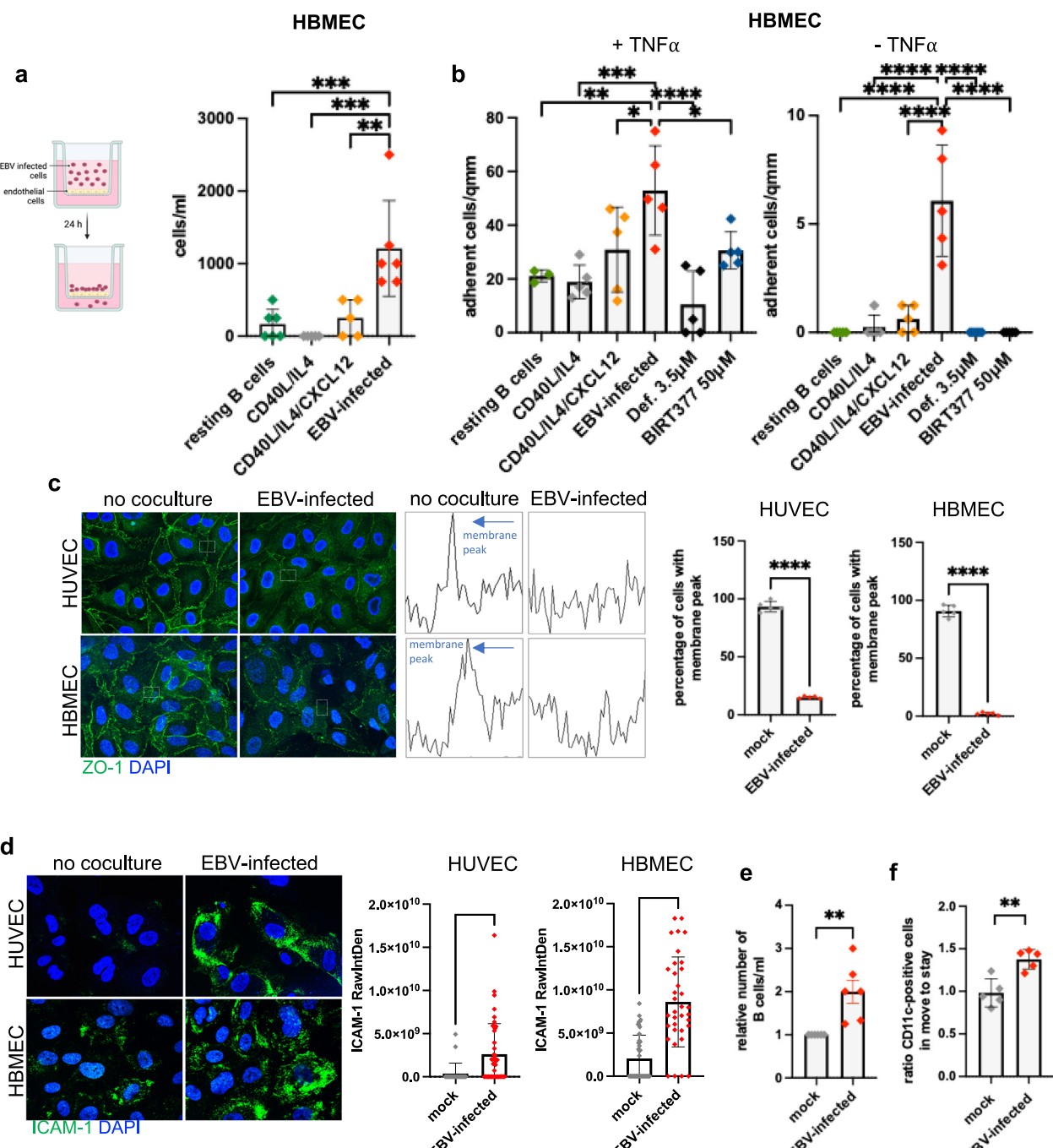

**Fig. 5 | EBV-infected B cells undergo diapedesis. a** Spontaneous diapedesis of primary resting B cells, B cells stimulated with CD40L and IL-4 or CD40L, IL-4 and CXCL12, and of EBV-infected B cells through a layer of human brain microvascular endothelial cells (HBMEC) cells. Bar graphs give the mean and standard deviation of cells that had migrated after 24 h of culture (*n* = 6 independent B cell samples). Created in BioRender. Poirey, R. (2025) https://BioRender.com/qnmhdf1. **b** We determined the ability of various B cell populations (see **a**)) to adhere to HBMEC cells under constant flow conditions, with or without prior stimulation with TNF-α. The graph shows the concentration of cells bound to the endothelial layer achieved by the different B cell populations. The effect of defactinib (Def.) (3.5 μM) or of an LFA-1 inhibitor (BIRT377 50 μM) on EBV-infected B cell adhesion is also described. Bar graphs give the mean and standard deviation (*n* = 5 independent B cell samples). **c** HBMEC and HUVEC endothelial cells were cocultured with EBV-infected B cells for 30 min. The effect of coculture on ZO-1 expression was evaluated 24 h later by immunofluorescence. Its localization was determined by intensity profiling of gray values, membrane peaks being marked with arrows (middle panel). Bar graphs give the percentage and the standard deviation of cells that have a membrane peak

(*n* = 5 independent B cell samples). **d** Same as in (**c**), but cells were analyzed for ICAM-1 expression. The bar graphs give the raw integrated density of fluorescence (Raw/Int/Den) in the different samples (*n* = 5 independent B cell samples, graphs showing mean and standard deviation). **e** Primary B cells labeled with a green fluorescent dye were mixed or not with EBV-infected B cells and placed above an endothelial cell layer. The relative number of primary B cells that crossed the endothelial barrier under both conditions after 24 h is given in the bar graph as mean with standard deviation (*n* = 6 independent B cell samples). **f** Same experiment as described in (**e**), but the stay and move populations were stained for CD11c. The relative proportion of CD11c+ cells in these two subsets was determined (*n* = 5 independent B cell samples, 1000 cells recorded). The graph shows the mean of these proportions with their standard deviation in cell populations that underwent coculture with EBV-infected B cells. The results are given relative to the CD11c+ proportions in the absence of coculture. Statistical significance was determined using one-way analysis of variance in (**a**) and (**b**) and two-sided paired *t* tests in (**c**), (**e**), and (**f**) and unpaired *t* test in (**d**). \**p* < 0.05, \*\**p* < 0.01, \*\*\**p* < 0.001 and \*\*\*\**p* < 0.0001. Source data are provided as a Source Data file.

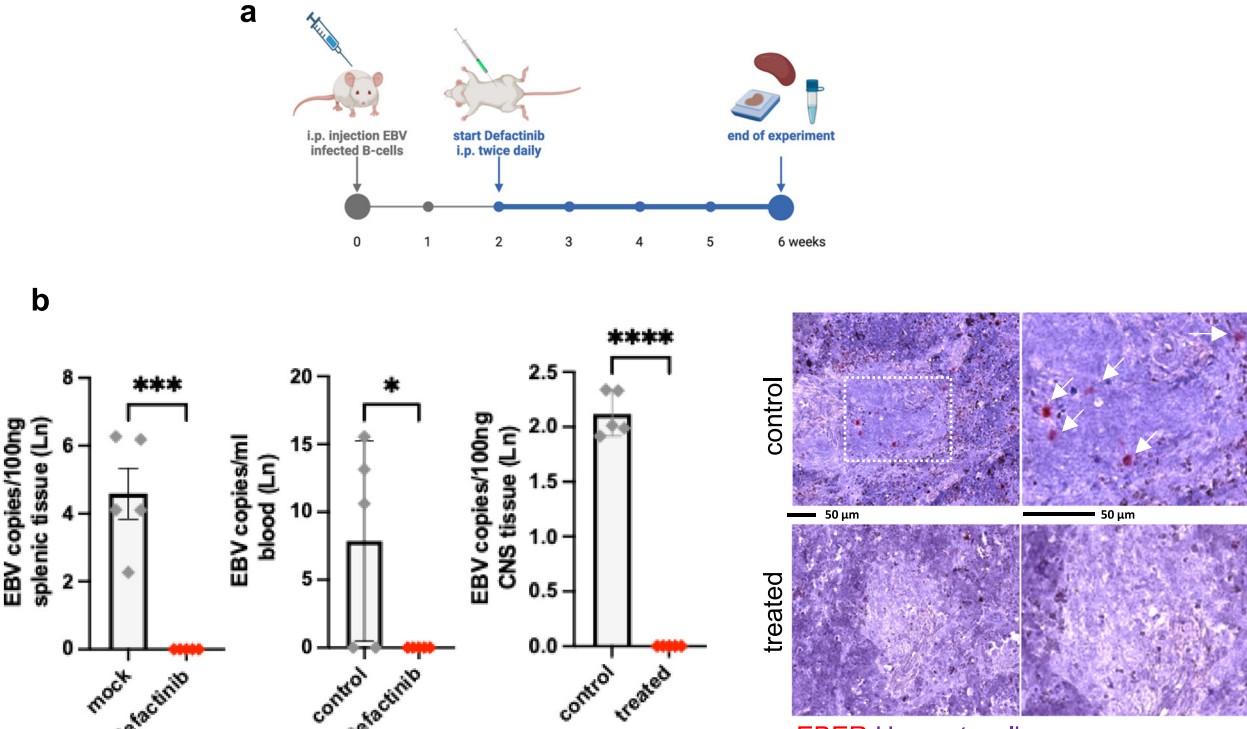

**Fig. 6 | Defactinib treatment protects immunosuppressed mice from splenic invasion by EBV-infected B cells. a** NSG mice were injected with primary B cells coated with low doses EBV. Two weeks later they were treated for 4 weeks with defactinib intra-peritoneally twice daily at a dose of 15 mg/kg. Untreated mice served as a negative control. Created in BioRender. Poirey, R. (2025) https://BioRender.com/me9d7ci. **b** Spleen tissue sections showed nodular infiltration by EBER-positive cells (red, marked with white arrows) in the control mice (upper panel, $n = 5$). In contrast, spleens from treated mice were completely devoid of EBV-infected B cells (lower panel, $n = 5$). Spleen tissue or peripheral blood after completion of the experiment were subjected to an EBV-specific PCR. Tissue sections from the brain and meninges of investigated mice were similarly analyzed for the presence of EBV sequences. The bar graphs give the mean EBV copies per 100 ng of DNA extracted from tissues after logarithmic transformation and their standard deviation. Statistical significance was determined using two-sided unpaired $t$ tests. ***$p < 0.001$; *$p < 0.05$; ****$p < 0.0001$. Source data are provided as a Source Data file.

initiate cell growth[1]. Previous work showed that LMP1 can induce CCL4 release and that CCR1 surface expression in EBV-positive Burkitt's lymphomas requires full latent gene expression[21,52]. We confirm that CCR1 and CCL4 expression in Burkitt's lymphoma cells requires full latent gene expression. Furthermore, we now show that both EBNA2 and LMP1 induce CCL4 release and that LMP1, but not EBNA2, induces CCR1 surface expression in primary B cells. These findings highlight how EBV latent genes influence the biology of their target. In particular, it shows that LMP1's functions extend beyond a mere induction of proliferation, and includes an activation of cell migration and diapedesis.

Our data also revealed new potential therapeutic targets as the CCL4-CCR1-FAK2 pathway and its downstream effectors can be relatively easily targeted as shown with the already available inhibitors. We found here in particular that defactinib, a molecule that has been extensively tested in clinical trials for multiple malignancies with variable efficacy, but with a good safety profile is able to eliminate infected B cells and prevent migration in the spleen in an animal model[27,28,53]. In order to parallel the clinical conditions as closely as possible, we used these inhibitors in vitro and in vivo at concentrations measured in the blood of treated patients. Notably, the combination of one CCR1 inhibitor previously used in clinical trials with very low doses of defactinib was able to block cell growth in vitro very efficiently. These two concentrations can probably be reached safely in patients with EBV-associated diseases. Previous clinical testing of the CCR1 inhibitors has shown their excellent safety profiles[54].

EBV-infected B cells' ability to induce ICAM-1 expression at the surface of endothelial cells, independently of an external

chemotactic stimulus, and to disrupt confluent endothelial tissues is likely to have pathogenic consequences. PTLD are frequently extra-nodal in location and involve organs such as the liver, the gastrointestinal tract or the central nervous system[7,8,10]. The ability of EBV-infected B cells to disorganize and cross endothelial barriers could explain their propensity to enter non-lymphoid tissues. ICAM-1 expression at the surface of endothelial cells is a crucial step in diapedesis and is likely to facilitate specific or non-specific transcytosis of inflammatory cells as we showed for resting primary B cells, in particular if the endothelium structure has been simultaneously damaged by EBV-infected cells. In particular, we found that EBV-infected B cells recruit and facilitate diapedesis of CD52$^{high}$CD11c$^+$ B cells, a process that is EBV-encoded IL-10-dependent. This could potentially explain why EBV-infected B cells and primary B cells, in particular T-bet$^+$CD11c$^+$ B cells infiltrate the central nervous system of patients with multiple sclerosis[34].

## Methods

### Ethics statement

This research complies with all relevant ethical regulations. The Ethics Committee of the University of Heidelberg approved all uses of human material (approval S/752/2022). All recruited volunteers provided written informed consent.

### B cell isolation

Peripheral blood CD19$^+$ B cells were isolated from fresh buffy coats by Ficoll density gradient followed by selection with anti-CD19 PanB Dynabeads and beads detachment, as recommended by the manufacturer (#11143D, Invitrogen, Germany).

## B cell stimulation

Peripheral blood CD19+ B cells were cultured in RPMI-1640 medium (#11875093, ThermoFisherScientific, Germany) supplemented with 10% fetal calf serum (FCS) and stimulated with 20 ng/ml human recombinant IL-4 (#A42602, Life Technologies, Germany) and 50 ng/ml soluble CD40L (#PPT-310-02-100, Peptrotech, Germany) (designated as CD40L/IL4) over 5 days and subjected to subsequent assays. In other experiments we used cells that were additionally stimulated with CXCL12 (CD40L/IL4/CXCL12). For stimulation with CXCL12 (#300-28 A, Peptrotech, Germany) 100 ng/ml of the chemokine were added to 4 days old CD40L/IL4 stimulated B cells and analyzed 24 h later.

## Viruses

Infections were performed with wild-type M81 or a mutant with a deletion in the BCRF1 gene, the EBV-encoded Il-10 (B828)[55]. We constructed the BCRF1 knockout mutant by deleting the BCRF1 gene locus from the TATA box to the end of the BCRF1 ORF (strain M81, KF373730.1 coordinates 9594 to 10183), using homologous recombination with the M81 bacmid. The deletion was obtained by exchanging this segment of the EBV genome with a kanamycin cassette cloned in the pCP15 plasmid. This was achieved by PCR amplification of the kanamycin cassette (primers see Supplementary Table 1).

We also constructed an EBV LMP1null virus (B738) by exchanging the LMP1 sequence (EBV coordinates 167717-168892) with a kanamycin resistance cassette by homologous recombination in the M81 wildtype BAC (KF373730.1). This cassette was amplified from the pCP15 plasmid by PCR (primers see Supplementary Table 1). After successful recombination, the kanamycin resistance gene was excised from the LMP1null virus using the FLP recombinase (pCP20) as described before[56].

## Virus production

For recombinant virus production, lytic replication of 293 cells containing the EBV recombinant virus (M81, EBV LMP1null, EBV-encoded Il-10null) were induced by transfection of a BZLF1 expression plasmid with co-transfection of a BALF4 expression plasmid[57]. The supernatants were collected 4 days post-transfection and filtered through a 0.45-µm filter to remove cell debris.

## Infections

Peripheral blood CD19+ B were exposed to various viruses to generate new virus-transformed cell lines (designated as EBV-infected cells) and were routinely cultured in RPMI-1640 medium supplemented with 10% fetal calf serum.

## Endothelial cells

Primary Human Brain Microvascular Endothelial Cells (HBMEC) were purchased from Innoprot (P10361, Innoprot, Spain) and cultured on fibronectin (2 µg/cm$^2$, #354008, Corning, USA) coated 75 cm$^2$ flasks in Endothelial basal medium supplemented with FCS, endothelial cell growth supplement and penicillin/streptomycin solution (#P60104, Innoprot, Spain). Human Umbilical Vein Endothelial Cells (HUVEC) and Human Dermal Microvascular Endothelial Cells (HDMEC, isolated from the dermis of juvenile foreskin and adult skin) were purchased from Promocell (C-12200 and C-12212, Promocell, Germany). HUVEC cells were grown in Endothelial Cell Growth Medium, HDMEC cells in Endothelial Cell Growth Medium MV (C-22110, Promocell, Germany). EBV-infected cells were cultured in RPMI-1640 (ThermoFisherScientific, Germany) and 10% FCS. For coculture experiments EBV-infected cells were kept in media required by endothelial cells.

## Cell lines and transfection

The EBV-negative Burkitt lymphoma cell line BL41 (kind gift from Gilbert Lenoir, IARC, France) and the EBV-positive MUTU cell lines (clones I and III) (kind gift from A. Rickinson, CRUK, Birmingham, UK) were cultured in RPMI-1640 and 10% FCS. BL41 was electroporated using

10 µL Neon tips with the following settings: 1350 V, 20 ms, 1 pulse. CD40L + IL-4 stimulated B cells were electroporated using 10 µL Neon tips with the following settings: 1900 V, 20 ms, 1 pulse. For transfection the Neon MPK5000 transfection system was used (#10431915, ThermoFisherScientific, Germany)

## Chemokines and neutralizing antibodies

Chemokines CCL3 (#300-08), CCL4 (#300-09), CCL5 (#300-06), hIL-10 (#AF-200-10) were purchased from Peprotech (Peptrotech, Germany), vIL-10 was purchased from Biotechne (#915-VL-010, Biotechne, Germany) and used at concentrations of 10–100 ng/ml. Neutralizing antibody against CCL4 was purchased from R&D (#MAB271, clone 24006, R&D, USA) and added at concentrations of 0.5–1.0 µg/ml 30 min prior to time lapse experiments.

## Inhibitors

In some experiments we added inhibitors 30 min prior to our analysis. These inhibitors were BX471 (S7604, Selleckchem, USA) at 1-10 µM, defactinib (S7654, VS-6063, Selleckchem, USA) at 0.5-10 µM, Latrunculin A at 5 µM (10010630, Cayman Chemical, USA), BDM (2,3-Butanedione-2-monoxime, 20828, Cayman Chemical, USA) at 50 mM, Blebbistatin (24169, Cayman Chemical, USA) at 50 µM, Y27632 (ROCK Inhibitor, 10005583, Cayman Chemical, USA) at 5 µM, ZCL278 (CDC42 Inhibitor, 14849, Cayman Chemical, USA) at 50 µM, LFA-1 Inhibitor BIRT377 (30112547, ThermoFisherScientific, Germany) at 50 µM. We also used AMD3100, a CXCR4 inhibitor at a 1 µM concentration (HY-10046, Medchem, Germany).

## ELISA

CCL3 (DY270), CCL4 (DY271), CCL5 (DY278) and human IL-10 (DY217B) were quantified by ELISA (DuoSet ELISA, R&D) in the supernatants of $1 \times 10E6$ EBV infected B-cells, of B cells stimulated with CD40L/IL4 or of unstimulated B cells that were cultured in 1 ml of RPMI/10%FCS for 24 h.

## Western blotting

Western blotting was performed as described before[8]. In brief, protein extracts were generated by resuspending cells in extraction buffer (50 mM Tris at pH 7.5/150 mM NaCl/0.1% SDS/1% sodium deoxycholate/1% Triton X-100) and sonication. After heat denaturation, extracts were loaded onto 10% or 12% SDS acrylamide gels. After electrophoresis, proteins were electroblotted onto an ECL-Membrane (GE10600001, Amersham Pharmacia) for 90 min at 25 V. After pre-incubation of the blot in PBS/5% dry milk powder containing 0.1% Tween 20, antibodies were added overnight at 4 °C. After washings in PBS, the blots were incubated for 1 h with horseradish peroxidase-coupled secondary antibodies (goat anti-mouse, w402b, Promega, final dilution 1:20000). Bound Abs were revealed by using the ECL detection reagent (#1705061, BioRad, Germany). We used antibodies against FAK2 (ab32571, clone YE353, 1:1000, Abcam), pFAK2 (Y402) (#592918, clone MAB6210, 1:1000, Cell Signaling), tubulin (T6557, clone GTU-88, 1:5000, SIGMA) and CCR1 (#53504, clone MAB145, 1:1000 R&D).

## Immunofluorescence

For studies of suspension cells (EBV-infected and controls), cells were pre-fixed in 1% paraformaldehyde (PFA) for 5 min. PFA-pre-fixed cells were dropped and dried on glass slides, fixed with 4% paraformaldehyde for 5 min at room temperature and permeabilized in phosphate-buffered saline (PBS) or tris buffered saline (TBS) 0.5% Triton X-100 for 2 min. Adherent cells were fixed in 4% PFA for 10 min and permeabilized as described for suspension cells. Fixed cells were incubated with the first antibody at 4 °C overnight, washed in PBS thrice, and incubated at 37 °C for 30 min with the secondary antibody and counterstained with DAPI and a blue or red membrane dye (BOT-30023,

CellBrite Cytoplasmic Membrane Dye, Biomol, Germany) to visualize cell membranes and lamellipodia. Slides were embedded in 90% glycerol and visualized with a confocal Olympus Fluoview 1000 microscope. We used antibodies against CDC42 (ab187643, clone EPR15620, 1:100 Abcam, UK), pPKCzeta (Thr410) (PA5-104967, polyclonal, 1:100, Invitrogen, Germany), ZO-1 (#33-9100, clone 1A12, 1:1000, Invitrogen, Germany), ICAM-1(#BMS1011, clone R6.5., 1:1000, ThermoFisherScientific, Germany), gm130 (#12480, clone D6B1, 1:100, Cell Signalling, USA), FAK1 (#3285, polyclonal, 1:1000, Cell Signalling, USA) and FAK2 (ab32571, clone YE353, 1:250, Abcam, UK). For stainings of ZO-1 and ICAM-1 after coculture, endothelial cells were seeded into a 18 Well µ-Slide at a concentration of $3 \times 10E5$ cells/ml and grown until confluency. $2 \times 10E4$ EBV-infected cells or control cells were added and incubated as cocultures for 30 min at 37 °C 5%CO$_2$ and washed off thereafter. 24 h after coculture cells were stained for ZO-1 or ICAM-1, respectively. We used the same algorithm to analyze intercellular spaces that appear after coculture and stained endothelial cells with Rhodamin-Phalloidin (A22287, Invitrogen, Germany).

## Live confocal microscopy
$1 \times 10E5$ EBV-infected cells or controls were stained with MitoTracker green or LysoTracker green (#M7514 and #L7526, ThermoFisherScientific, Germany) according to the manufacturer's protocol. We used a concentration of 200 nM for MitoTracker labeling. Cells were seeded into ibidi µ-Slide 18 Well Glass Bottom (#81816, Ibidi, Germany) in RPMI-1640 and 10% FCS and visualized using a confocal microscope (Olympus FluoView FV1000). We used live cell probes (SIR-actin and SIR-tubulin) for actin and tubulin staining and imaging in living cells. Both probes were purchased from Spirochrome (#CY-SC001 and #CY-SC002, Spirochrome, Switzerland) and used as to the manufacturer's instructions with an incubation time of 60 min at a 1 µM final concentration. For centrin, $7 \times 10E5$ EBV-infected cells were transfected with the pEGFP-centrin-1 plasmid (#72641, Addgene, USA) using Neon Transfection system (#10431915, ThermoFisherScientific, Germany). For each sample, we electroporated one time using 10 µL Neon tips with the following settings: 1350 V, 30 ms. One day after transfected cells were analyzed with a confocal microscope (Olympus FluoView FV1000).

## Scanning electron microscopy
Cells grown in suspension were fixed with cacodylate-buffered aldehyde (2% freshly prepared Formaldehyde plus 2% glutaraldehyde), post-fixed with 1% buffered OsO$_4$, dehydrated in graded steps of ethanol following critical point drying (Balzers 030) using porous pods (Baltic preparation, Wetter, Germany) as containers. The dried cells were shed onto gluey carbon tabs (science services, Munich, Germany) for mounting on standard Al-subs and sputter coated with Au/Pd 80:20 (Batic preparation, Wetter, Germany). Micrographs were taken with a Zeiss Auriga SEM (Carl Zeiss Oberkochen, Germany) at 2 kV acceleration Voltage and about 2 mm work distance using an inlens detector for secondary electrons.

## Time lapse microscopy
For time lapse experiments, $3 \times 10E5$ cells/ml (EBV-infected high) or $3 \times 10E4$ cells/ml (EBV-infected low) were seeded on Ibidi chemotaxis slides (#13478749, Ibidi GmbH, Germany) into a bovine collagen matrix with a concentration of 1.5 mg/ml bovine collagen I (#A106444-01, ThermoFisherScientific, Germany). The slide was inserted into a 37 °C heating and incubation system for the whole duration time lapse analysis. Images were acquired every minute over a period of 15 min using a confocal Olympus Fluoview 1000 microscope. Manual tracking was performed using the Fiji Tracking tool and presented as an overlay of dots and lines for 2D tracks. Directionalities and velocities from manual trackings were calculated using Ibidi chemotaxis and migration tool. We generated aligned 2D trajectory plots ("spider plots") by setting all

(x,y) coordinates of the cells' starting points to (0,0). The data were statistically analyzed by Rayleigh test using the Ibidi chemotaxis and migration tool. We plotted the mean cell displacement by the square root of time to generate square root time profiles.

## Adhesion assay under flow conditions
BMEC or HUVEC cells were seeded at a concentration of $(3-5) \times 10E5$ cells/ml onto ibidi µ-Slides VI 0.4 (#80606, ibidi, Germany) and cultured until confluency. Slides were precoated with 100µg/ml with fibronectin (2 µg/cm$^2$, #354008, Corning, USA) upon seeding. Slides were placed in a thermostated hood (37 °C) and time lapse microscopy was performed using a confocal microscope (Olympus FluoView FV1000). Pictures were taken every 10 seconds over a period of 15 min. µ-slides were connected to the ibidi Pump system (ibidi, Gräfelfing, Germany) and exposed to various cell suspensions (EBV-infected B cells, stimulated and resting B cells) at a concentration of $1 \times 10E6$/ml. After initial perfusion of the flow chamber at 0.6 dynes/cm$^2$ for 2 min for equilibration, the total cell suspension was perfused through the chamber at a constant flow rate (1.5 dynes/cm$^2$) and images recorded using a time lapse recording system connected to the microscope (Olympus FluoView FV1000). After 10 min of perfusion, the flow rate of the cell suspension was raised so that wall shear stress increased from 1.5 to 3.0 dynes/cm$^2$. Adherent leukocytes were identified and counted at 2-min intervals during the 10-min perfusion at 1.5 dynes/cm$^2$ as previously described[58]. In some experiments the cells were cultured in LFA-1 Inhibitor BIRT377 50 µM or defactinib 3.5 µM 30 min prior to the experiment. For some experiments BMEC and HUVEC cells were stimulated with TNF-α at a concentration of 100 ng/ml 24 h (#300-01 A, Peptrotech, Germany) prior to the experiment.

## Transwell migration assays with endothelial barrier
$2 \times 10E4$ cells of EBV-infected and/or uninfected CD19+ B cells were seeded into the upper compartment of transwell insets with an area of 0,32 cm$^2$ and 3 µm pore size. Chemokines CCL3, CCL4, CCL5, hIL-10 and vIL-10 were added at different concentrations (10-100 ng/ml) into the lower compartment and the number of migrating cells determined after 1 to 24 h. In experiment where EBV-infected cells served as attractant for B cells, B cells were labeled with CytoTrace Green CMFDA (#22017, AAT Bioquest Inc., USA) and seeded into the upper compartment of the transwell inset. EBV-infected cells were seeded at a concentration of $2 \times 10E4$ per ml into the lower compartment and numbers of migrating cells determined after 24 h. In other experiments endothelial cells HUVEC, BMEC or HDMEC were seeded at a concentration of $3 \times 10E5$/ml into the upper compartment of the inset and cultured until confluency (2–3 days). 2–3 days after seeding EBV-infected cells were seeded into the upper compartment and migration determined after 24 h. In some experiments B cells and EBV-infected cells were seeded simultaneously into the upper compartment of the inset. In this case B cells were labeled with CytoTrace Green CMFDA (#22017, AAT Bioquest Inc., USA) and the number of migrating cells determined after 24 h.

## TEER measurements
HBMEC or HUVEC cells were seeded as described above for transwell migration assays with endothelial barrier. Trans-endothelial electrical resistance (TEER) was measured until stable values were reached[59]. For coculturing, insets were placed upside down and $5 \times 10E5$ EBV-infected cells applied onto the basolateral side of endothelial cells in a volume of 50 µl of endothelial media. Cells were left for 30 min and gently washed off. TEER was measured the subsequent day and given as TEER$_{REPORTED}$ relative to the day before coculture was performed. For HUVEC cells barriers were induced by adding DBcAMP 250µM (Dibutryl cAMP, #HY-B0764G, Medchem, Germany) one day before contact with EBV-infected cells.

## Chemotaxis assay using Boyden chambers

The chemotaxis assays were performed using blind well chemotaxis chambers (Neuro Probe, #BW100), with compartments separated by a 5 µm polycarbonate filter (Neuro Probe, #PFA8). The lower compartments were filled with either 10%FCS/RPMI or chemokines (CCL3, CCL4, CCL5, human and/or EBV-encoded IL-10) in 10%FCS/RPMI. The upper compartment was filled with 100 µl 10%FCS/RPMI containing $5 \times 10^4$ cells. The chambers were incubated for 1 h at 37 °C/5% $CO_2$ and afterwards the cells in the lower compartment were counted.

## Flow cytometry

Cells were washed and resuspended in 100 µL of PBS and 0.1% BSA incubated with primary antibodies against CD11c conjugated to APC (# 337208, clone Bu15, 1:20, Biolegend, USA), CD52 conjugated to APC (#318904, clone QA19A22, 1:20, Biolegend, USA) or ICAM-1 conjugated to PE (#MCA1615PE, clone 15.2, 1:20, Biorad, Germany) for 30 min on ice. After washing, cells were resuspended in 200 µL of PBS and 0.1% BSA and analyzed using a BD FACS calibur (Becton, Dickinson and Company). In coculture experiments or experiments were EBV-infected B cells were used to attract B cells, these cells were labeled with CytoTrace Green CMFDA (#22017, AAT Bioquest Inc., USA), 1000 (CD11c), 2000 (CD52) or 10000 (ICAM-1) cells were recorded. Using FSC and SSC gating dead cells were excluded and in case of coculture experiments a second gating was used for green cells and the percentage/MFI of these cells were reported (CD11c and CD52) Post-acquisition analysis was performed using the FlowJo Software (Becton, Dickinson and Company). Gating strategies are shown in Supplementary Fig. 11.

## Calcium signaling

HUVEC, BMEC or HDMEC were seeded at a concentration of $3 \times 10E5/ml$ into ibidi µ-Slide 18 Well Glass Bottom (Ibidi, Germany) and cultured until confluency. Cells were labeled with FLUO4-AM (#F14201, ThermoFisherScientific, Germany) at a final concentration of 2.5 µM. Before dilution into the loading medium equal volume of 20% (w/v) Pluronic in DMSO (P3000MP, ThermoFisherScientific, Germany) was added. Cells were incubated with the AM ester for 60 min at 37 °C. Before fluorescence measurements, cells were washed in indicator-free medium and then incubated for a further 15 min to allow complete de-esterification of intracellular AM esters. Baseline pictures were acquired and $2 \times 10^4$ EBV-infected or control cells were added and changes in signals detected using a confocal microscope (Olympus FluoView FV1000). Mean fluorescent intensities were determined using Fiji image software[60].

## Single-cell RNA sequencing[61–63]

$2 \times 10E4$ uninfected CD19+ B cells were labeled with CytoTrace Green CMFDA (#22017, AAT Bioquest Inc., USA) and seeded into the upper compartment of a transwell insets with an area of 0.32 cm² and 3 µM pore size in 200 µl of RPMI-1640 and 10% FCS. $2 \times 10E4$ unlabeled EBV-infected cells were added to the lower compartment in 500 µl of RPMI-1640 and 10% FCS. Cells were incubated for 24 h and green labeled B cells of the upper (designated as "stay") and lower (designated as "move") compartment each sorted into two 384 well plates by Fluorescence Activated Cell Sorting using a BD FACSAria (Becton, Dickinson and Company). Cells were subjected to SMART-seq 2 scRNA-seq platform. Transcript reverse transcription and amplification were performed following the protocol of Smart-seq2. Libraries were constructed with the Nextera XT DNA Library Prep kit (Illumina, San Diego, CA) and sequenced on Novaseq paired end 100pbSP (Illumina, USA).

For Creation of the Seurat object and normalization raw FASTQ files (R1: 101 base pairs, R2: 101 base pairs) were aligned using STAR on the GRCh38 human reference genome with the soloType function "SmartSeq" (v2.7). Quality controls were used to exclude cells with a number of detected genes below 3500 and above 5500 and 6300 (move and stay conditions, respectively) and cells with more than 15% of transcripts encoded by the mitochondrial genome. The resulting count matrices were log-normalized using Seurat NormalizeData with a scale factor of 10.000 and a scaling step was performed using Seurat ScaleData with all genes from the matrices as features to cell cycle calculation. Each filtered count matrix was normalized a second time with Seurat SCTransform (vst.flavor = "v2") and dimension reduction was performed with RunPCA (npcs = 5).

For Metadata creation cell phenotype annotations were identified using singleR and Celldex R packages against cell markers from the MonacoImmuneData database. All calculations were made from the "SCT" assay of the Seurat merged object.

For Data merging, we merged the condition-specific objects (move and stay) using Seurat v4.3. Variable features from scale.data slot were used to center and reduce the merged object with RunPCA and embedded in two dimensions with RunUMAP, excluding BCR- and TCR-encoding genes [Supp. Ref 2,3] from the lists of variable genes (regex: IG[HKL][VDJ] |IGHG[1-4]|IGH[MDE] |IGKC|IGLL|IGLC[1-7] | IGHA[1-2] |TR[ABGD][CV]) determined by the Seurat function VariableFeatures.

Louvain clustering was performed with the FindClusters function, with a resolution of 0.6. To annotate each cluster, we ran a 'one-versus-all' Differential Expression Analysis (DEA) for each cluster (Seurat, FindAllMarkers, Wilcoxon rank-sum test), keeping only upregulated genes with a avg_log2FC > 0.8, pct.1 > 0.6 and a Bonferroni-adjusted P value < 0.001. The resulting list of genes was used as input to the 'enrichGO' function of clusterProfiler package (v4.12.0, parameters: ont = "BP", OrgDb = org.Hs.eg.db, keyType = "SYMBOL" and pvalue-Cutoff = 0.05). We then removed redundancy in the output list of GO terms with the 'simplify' function with a cutoff of 0.70. For the annotation of cluster 2, all the upregulated genes from the differential expression analysis were considered (logfc.threshold = 0.25).

At the single-cell level, enriched pathway visualization was performed using the DEenrichRplot function of the Seurat R package. The maximum number of genes to perform the enrichment calculation was set to 500 and only pathways with FDR < 0.05 and logfc.threshold = 0.6 were kept, applying GO_Biological_Process_2023 database.

Single-cell differential gene expression list between move and stay conditions was calculated using Seurat FindMarkers (assay = "SCT") from merged matrix. The LogFC threshold was set to 0.6 with p_val_adj below 0.05. All B cells were retained for testing.

## CRISPR-Cas9 cloning for CCR1, CCL4 and hIL-10 null mutants

The pU6-(BbsI)-CBh-Cas9-T2A-mcherry-P2A-Ad4E4orf6 plasmid (Add gene #64222) was digested with XhoI and EcoRI to remove the Ad4E4orf6 cassette and a stop codon was inserted through primer annealing (Supplementary Table 1) and ligation. Subsequentially, the modified plasmid was digested with BpiI and the guide DNA sequences were introduced using primer annealing (Supplementary Table 1) and ligation. The correctness of the cloning was verified via sequencing using the primer listed in Supplementary Table 1. The design of the guide DNA was performed using publicly available tools: Chopchop (https://chopchop.cbu.uib.no/) and the IDT design tool (https://eu.idtdna.com/site/order/designtool/index/CRISPR_SEQUENCE). The gDNA sequences were chosen based on the on- and off-target scores predicted by the different tools.

## Cellular knockout construction

CCL4 and CCR1 plasmids were transfected into cells infected with EBV WT (M81), hIL-10 plasmids were transfected into an EBV-encoded IL-10 deficient mutant of the M81 virus (B828). Transfection was performed using Neon Transfection system (ThermoFisherScientific, Germany). For each sample, we electroporated one time using 10 µL Neon tips with the following settings: 1350 V, 20 ms, 2 pulses. One day after transfected cells were sorted for mCherry by Fluorescence Activated

Cell Sorting. Cells were cultured in RPMI-1640 and 10%FCS (CCR1[null]) and supplemented with 100 ng/ml CCL4 (CCL4[null]) or 10 ng/ml Il-10 (human IL-10[null]) and used for transwell or time lapse microscopy assays.

### Animal model of EBV infection

$1 \times 10E6$ B cells were infected with the M81 Virus at an MOI of 3 per cell and injected into 5-week-old male NSG mice (NOD.Cg-*Prkdc*[scid] *Il2rg*[tm1Wjl], DKFZ in house breeding). Defactinib or carrier solution (50% PG300, 5% Tween20, 40% H20, 5% DMSO) was given i.p. at a dosage of 15 mg/kg twice daily starting 14 days after infected cells were given. The infected mice ($n = 5$) were monitored for 6 weeks post-infection and then euthanized by $CO_2$ inhalation. Animals were housed in an BSL2 SPF facility. Control mice ($n = 5$) were hosted separately from treated animals. Animal experiments were approved by the Regierungspräsidium Karlsruhe (G-160/22) and are compliant with the institutional laboratory animal research guidelines. All efforts were made to minimize animal suffering and to reduce the number of animals used. Mice were maintained in a specific-pathogen-free, standardized environment with $22 \pm 2\,°C$ temperature, $55 \pm 10\%$ humidity, 12 h light/dark cycles and fed with a standard diet according to the German Cancer Research Center guidelines.

### DNA extraction and EBV copy number of spleen tissue[64]

Purification of total DNA from fixed, paraffin-embedded spleen tissue was carried out with DNeasy Blood & Tissue kits (#69504, QIAGEN, Germany). Briefly, paraffin was removed by extraction with xylene. Spin-Columns were used to isolate total DNA. EBV BALF5 gene locus was amplified from total DNA by qPCR of 100 ng tissue DNA. EBV copy number was calculated using a standard curve.

### Immunohistochemistry and EBER in situ hybridization

Mice organs were fixed in 10% formalin overnight and embedded in paraffin blocks. Three-micrometer-thick continuous sections were prepared. The presence of EBV was detected by in situ hybridization with an EBER-specific peptide nucleic acid probe, in conjunction with a PNA detection kit (K5201, Dako, USA) following the manufacturer's protocol. In parallel, adjacent sections were stained with H&E. Images were taken with an Aperio Digital Pathology Slide Scanner (Leica, Germany) and analyzed using QuPath Software[65].

### Statistical analysis

GraphPad Prism 9 was used to conduct all statistical analysis. The error bars represent the standard deviation of the data sets. Statistical significance was determined using Student's *t* test or ANOVA analyses combined with Dunnett's multiple comparisons test. Bar graphs include means and their standard deviations. P values are displayed by asterisks with $*p < 0.05$, $**p < 0.01$, $***p < 0.001$ and $****p < 0.0001$.

### Dose-response analysis and Isobologram

Dose response analysis was performed by first fitting a four-parameter log-logistic function to both defactinib and BX471. Isobolograms were computed by first determining the average observed result at several substance mixtures, and then using the available dose–response fits to identify the corresponding doses required by the component doses alone. All analyses were performed using R Version 4.3.0 and the DRC package[66] as well as the Graphpad Prism software.

### Reporting summary

Further information on research design is available in the Nature Portfolio Reporting Summary linked to this article.

## Data availability

The scRNAseq data generated in this study have been deposited in the ncbi database under accession code Bioproject" PRJNA1255185. All other data are included in the Supplementary Information. The raw numbers for charts and graphs are available in the Source Data file whenever possible Source data are provided with this paper.

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

## Acknowledgements

The authors are very grateful to the members of the core facility imaging and microscopy at DKFZ. We are particularly indebted to Dr. Karsten Richter for his tremendous help and expertise with electron microscopy. We also thank the genomics core facility of the DKFZ and Dr. Ivo Buchhalter for their help with scRNAseq. The study was funded by DKFZ, DZIF, and Inserm. This project has partially received funding from the European Union's Horizon Europe Research and Innovation Actions under grant no. 101137235 (BEHIND-MS). Views and opinions expressed are however those of the author(s) only and do not necessarily reflect those of the European Union nor the granting authority. Neither the European Union nor the granting authority can be held responsible for them.

## Author contributions

S.D. and H.J.D. planned experiments, analyzed data and wrote the paper. S.D., F.B., A.S., D.J., and R.P. conducted experiments. C.D. and T.H.L. performed statistical analyses and migration modeling. M.Z. and P.S. conducted bioinformatic analyses.

## Funding

## Competing interests

The authors declare no competing interests.
