## [Transparent Peer Review file · Nature Communications]

Epstein-Barr virus induces aberrant B cell migration and diapedesis via FAK-dependent chemotaxis pathways

Corresponding Author: Professor Henri-Jacques Delecluse

Version 0:

Reviewer comments:

Reviewer #1

(Remarks to the Author)

The authors have reported that EBV-infected B cells secrete multiple chemokines, among which CCL4 shows CCR1-dependent cell migration of EBV-infected B cells. The EBV-infected B cells induced diapedesis of CD52- and CD11c-positive resting B cells via IL-10. Although the findings by the authors are potentially important, the majority of the presented data seem to relate to known chemokine responses and do not appear to elucidate the essential pathogenesis of autoimmune diseases caused by EBV infection.

Major concerns:

The data presented in this manuscript do not substantially advance the mechanistic understanding of pathogenesis of autoimmune diseases caused by EBV infection. Essentially, no mechanism has been presented on induction of chemokines by EBV-infected B cells. While much of the authors' data on CCL4-induced changes in cellular responses are convincing, they can be inferred from the well-known functions of chemokines. In this sense, the present data are mainly confirmative.

Other issues:

1. Lines 65-67: Is directional migration of B blasts induced by CXCL12 or is the differentiation state changed when primary B cells have been exposed to CX40L, IL-4, and CXCL12? To understand the roles of CXCL12 in B cell blast motility, experiments with AMD3100 should be added.
2. Lines 67-68: In this experiment, do the authors observe chemotaxis or chemokinesis?
3. Lines 88-90: Supplementary Figure 1 does not have data for IL-10.
The authors need to validate the expression levels of human and viral IL-10 in EBV-infected B cells.
4. Line 172: The term "viral IL-10" has not appeared in any of the previous texts, so some explanation should be added.
5. Lines 626-629: For multiple comparisons, the method of pos-hoc tests should be described as well as ANOVA. In addition, the range of P-values corresponding to the number of stars should be shown.
6. Half-width spaces between numbers and units are missing (for example, lines 110, 327, 834, 947, 1021, 1022, Fig. 3c and d, Suppl. Fig. 2b-e, Fig. 3c).

Reviewer #2

(Remarks to the Author)

Epstein-Barr virus (EBV) infection of human B cells can cause human lymphomas and is also associated with the development of autoimmune diseases, particularly multiple sclerosis (MS) and lupus. The mechanisms(s) by which EBV infection of B cells might increase the development of MS are not fully understood, but may involve both increased EBV-directed antibodies that cross-react with CNS antigens, as well as increased migration of EBV-infected B cells across the blood-brain barrier. In this paper, the authors present a wealth of data suggesting that EBV-infected B cells display features of homing cells, including an increased ability to migrate through endothelial cells. They show that increased migration is mediated by increased expression of both CCL4 and CCR1 in EBV infected B cells, which results in activation of the FAK2 kinase. Furthermore they show that a drug which inhibits the FAK1 and FAK2 kinases decreases migration of EBV-infected B cells into the CNS in a mouse model and also inhibits EBV-induced lymphomas in this model. Finally, they show that increased expression of EBV-encoded IL10 also helps to recruit uninfected primary B cells towards EBV-infected B cells.

Overall, the data are of high quality and the findings are significant and novel, suggesting that FAK kinase inhibitors might be useful for treating EBV-associated diseases, including MS.

However, the paper would be improved by including the following:

1. It is not clear how many donors were used to generate the EBV-infected B cells used in the study. Did they all come from a single donor? If so, the authors need to show that similar results are obtained in at least two different EBV-infected cell lines, obtained from different donors. They should also state how long after EBV infection the studies were performed. Is the increased migration phenotype only seen in early passage EBV-infected cell lines or is this a persistent phenotype?
2. The authors should make some attempt to identify the EBV protein(s) that mediate increased CCL4 and CCR1 expression in B cells. Does this only occur in B cells with "type III" latency or does it also occur in B cells with more stringent forms of viral latency (which are much more common in the human host). This should be possible since another group reported that CCR1 expression is increased in EBV-positive Burkitt lymphomas that have drifted to "type III" latency in vitro. At the very least they could over-express the EBV EBNA2 or LMP1 proteins in an EBV-negative or type I EBV+ Burkitt line to determine if EBNA2 or LMP1 expression mediates the phenotype they are describing.

Reviewer #3

(Remarks to the Author)

In this manuscript, Delecluse et al describe a process in which Epstein-Barr virus infected B cells upregulate chemokines and their receptors to support direct migration, recruitment of uninfected B cells, and diapedesis along with disruption of endothelial cell barrier function. Overall, this is a tour de force with many experiments characterizing the molecular details of this newly described molecular circuitry enabling a robust migratory phenotype in human B cells.

The initial findings are that EBV infection promotes a directed B-cell migration phenotype relative to CD40L/IL4 induced B blasts and uninfected B cells as evidenced by live cell microscopy and image analysis and similar in velocity and direction to B blasts treated with CXCL12. This activity is associated with upregulation of both the chemokines CCL3, 4, and 5 (MIP1a, b, and RANTES) and their cognate receptors CCR1. The authors demonstrated that EBV-infected B cells are recruited to CCL4 and that the depletion of CCL4 by antibody or gene knockout significantly reduces motility. Likewise, inhibition of CCR1 with BX471 and KO also suppresses motility. Evidence of polarized cell movement is captured by SEM and IF for cdc42 and phospho-PKCzeta as well as increased phosphorylation of FAK2. The EBV-infected cell motility requires CDC42 and ROCK activity as well as FAK2 as evidenced by inhibition with defactinib and combination experiments show potential evidence of synergy between CCR1 antagonism and FAK2 inhibition. Another fascinating finding is that EBV-infected cells recruit uninfected B cells by transwell assay. This activity is apparently driven by viral IL-10 as a knockout decreases and providing exogenous vIL-10 increases uninfected B cell movement across the transwell. Single cell sequencing characterizes the motile population as upregulated CD52, which is validated by flow cytometry. Further characterization indicates an increase in CD11c in this population.

The next set of experiments include demonstrating that EBV-infected B cells can undergo diapedesis through various endothelial cell barriers. This is very intriguing and supported by several experiments comparing to resting B cells, B blasts and CXCL12 directed motile B blasts, none of which act like EBV-infected B cells in terms of crossing the endothelial barrier or disrupting the trans-endothelial electrical resistance. These attributes and endothelial cell adhesion all rely on CCR1 and FAK2. Finally, EBV-infected B cell invasion of the spleens of NSG mice were dependent on FAK2.

Overall, there is a significant amount of data supporting the directed motility driven by CCR1/CCL4 and FAK2 dependent actin polymerization processes. Further, diapedesis by EBV-infected B cells and disruption of the TEER of endothelial cell barriers support a model whereby EBV-infected cells have the capacity to potentially breach the BBB in MS.

While most of the experiments were very well controlled and easily interpretable, there were a few that could have been either left out of the study (eg the IL10 recruitment of uninfected B cells) or bolstered with additional data (eg scRNAseq).

Minor experimental comments:

1. The scRNAseq of uninfected B cells provides an initial glimpse of what changes are driven by exposure to the EBV-infected cell supernatants and presumably a vIL-10 mediated process, but this is limited to CD52 RNA and validation by flow along with CD11c flow. It would be interesting to see what the overall changes are between motile and non-motile resting B cells.
2. In Figure 2c (and other similar expmts) are these B cell clusters moving or single cells?
3. Need isobologram to argue for synergy of defactinib with BX471
4. How many uninfected B cells are moving in the transwell assays? These are normalized and makes it difficult to interpret

Minor text and stylistic comments:

1. In Figure 1, it would be good to change the color of the CD40L/IL4/CXCL12 from that of CD40L/IL4 as it's difficult to distinguish the two.
2. It would be easier to follow if the time courses were described initially for the B blast and EBV infections to understand what the comparisons are.
3. Which band is the phospho-FAK2 in Fig 3c? Presumably the one with the asterisk, but usually an arrow would mark the actual band and an asterisk perhaps the background band above it that doesn't change.
4. The data presented in fig 5c and d should be quantified

5. Line 96 'CCL4'

6. Page 5, line 59 'very high' relative to what?

Version 1:

Reviewer comments:

Reviewer #1

(Remarks to the Author)

I understand the novelty of this study in the revised version. Although the data presented are primarily confirmatory regarding the well-known functions of chemokine and chemokine receptors, the experiments were accurately performed and are reliable. The potential molecular mechanism on induction of chemokines by EBV-infected B cells has now been included in the text and Figures.

minor issues:

1. Supplemental Fig. 1 (f) , line 25: "CXCR5" must be "CXCR4".
2. Supplemental Fig. 1 (f) , line 28: "life cell imaging" must be "live cell imaging".

Reviewer #2

(Remarks to the Author)

This revised paper has adequately addressed the points made by the three previous reviewers. The manuscript (which was already highly significant and original) is now even further improved.

Reviewer #3

(Remarks to the Author)

The authors have satisfactorily responded to my concerns. The manuscript is improved and in excellent shape for publication. This study should have a large and sustained impact on the EBV field.

Rebuttal letter, S. Delecluse et al.

We would like to thank the referees for their thorough evaluation of our paper. We are pleased to read that referee 2 and 3 found that the “findings are significant and novel”, and that the paper “is a tour de force with many experiments characterizing the molecular details of this newly described molecular circuitry enabling a robust migratory phenotype in human B cells”. While referee 1 finds that “the majority of the presented data seem to relate to known chemokine responses and do not appear to elucidate the essential pathogenesis of autoimmune diseases caused by EBV infection”, he/she adds that the findings on CCL4 “are potentially important”.

Reviewer #1 (Remarks to the Author):

The authors have reported that EBV-infected B cells secrete multiple chemokines, among which CCL4 shows CCR1-dependent cell migration of EBV-infected B cells. The EBV-infected B cells induced diapedesis of CD52- and CD11c-positive resting B cells via IL-10. Although the findings by the authors are potentially important, the majority of the presented data seem to relate to known chemokine responses and do not appear to elucidate the essential pathogenesis of autoimmune diseases caused by EBV infection.

Answer: We fully agree with this reviewer, our paper does not fully elucidate the role of EBV in the pathogenesis of autoimmune diseases. However, we have discovered that this virus activates important biological processes, migration and diapedesis, in the absence of external stimulation, and this has potential therapeutic consequences for autoimmune diseases, among others. We feel, and referees 2 and 3 seem to share this opinion, that this represents a significant advance in the field. Moreover, our paper highlights how EBV uses an unusual paracrine mechanism to induce migration. This results from simultaneous expression of the CCL4 chemokine and one of its receptors CCR1. Even if the molecules involved are well characterized, we think that this is not a trivial observation.

Major concerns:

1) The data presented in this manuscript do not substantially advance the mechanistic understanding of pathogenesis of autoimmune diseases caused by EBV infection.

Answer: We disagree with this comment, as seemingly do referees #2 and #3. We think that the observation that EBV infection induces B cell migration and diapedesis is novel and contributes to a better understanding of EBV pathogenic potential. Indeed, it can potentially explain the presence of EBV-infected cells in the brain of patients with multiple sclerosis and EBV-associated lymphomas of the central nervous system. Same remarks apply to the gut in which EBV-associated lymphomas surprisingly preferentially develop, as mentioned in the paper (see introduction for references).

2) Essentially, no mechanism has been presented on induction of chemokines by EBV-infected B cells.

Answer: We have used multiple experimental systems to answer this question. We first monitored CCL4 and CCR1 expression in Mutu I (latency I) and Mutu III (latency III) Burkitt's lymphoma clones. This assay showed that cells in latency III, but not those in latency I, strongly express both CCL4 and CCR1 (pages 6-7 lines 127-147 and Fig. 2 e-g and Suppl. Fig. 3d). We then transfected EBNA2 and LMP1, both in EBV-negative BL41 Burkitt's lymphoma cells and in primary B cells stimulated with CD40 ligand and IL-4. These assays revealed that both EBNA2 and LMP1 are potent CCL4 inducers. However, only LMP1, and not EBNA2, could induce CCR1 surface expression in either experimental system. To confirm LMP1's role in CCR1 and CCL4 expression, we infected primary B cells with a LMP1^{null} mutant (M81/ Δ LMP1). Relative to B cells infected with wild type viruses, B cells infected with M81/ Δ LMP1 released CCL4 and expressed CCR1 at 5- and 10-times lower levels than their wildtype counterparts, respectively. Altogether, we found that cells in latency III and cells that express LMP1, but not cells in latency I express both CCR1 and CCL4 (see Fig. 2 d to g).

pages 6-7 lines 127-147:

EBNA2 and LMP1 control CCR1 and CCL4 expression. Latently infected B cells proliferate under expression of the latent genes, among which the Epstein-Barr nuclear antigen 2, a transactivator that activates the Notch pathway and the latent membrane protein 1, a permanently active viral homolog of CD40, play a crucial role. To establish whether these viral genes control CCL4 and CCR1 expression in infected B cells, we first monitored their expression in an early and a late passage of the MUTU Burkitt's lymphoma cell line. While an early passage of this line (MUTU I) has a restricted latent protein expression pattern largely limited to EBNA1 (latency I), late passage cells (MUTU III) express all latent genes, including EBNA2 and LMP1 (latency III)¹⁷. CCR1 surface expression and CCL4 release in MUTU III was respectively 8 (CCR1) and 600 (CCL4) times higher than in MUTU I, suggesting that expression of these proteins is associated with latency III (Fig. 2e). To determine which latent gene is responsible for these effects, we transfected EBNA2 or LMP1 in CD40L+IL4 activated B blasts. These assays revealed that LMP1 is a potent inducer of CCL4 and CCR1 expression (Fig. 2f). Interestingly, EBNA2 also activated CCL4 release, but was unable to induce CCR1 expression (Fig. 2f). Similar results were obtained after transfection of the cell line BL41 with LMP1 or EBNA2 (Suppl. Fig. 3d). To confirm LMP1's role in this process, we infected primary B cells with a LMP1null mutant (M81/ Δ LMP1). Relative to B cells infected with wild type viruses, B cells infected with M81/ Δ LMP1 released CCL4 and expressed CCR1 at 10 and 5 times lower levels, respectively (Fig. 2g). This approach was unfortunately not possible for EBNA2 as the M81/ Δ EBNA2 mutant fails to initiate transformation. Altogether, we conclude that LMP1 and EBNA2 collaborate to initiate the chemokine loop that leads to migration.

Fig. 2

Fig. 2 Migration of EBV-infected B cells is driven by CCL4 and CCR1. ... e-g) EBV latent genes and CCL4-CCR1 expression. e) CCR1 surface expression on Burkitt cell line MUTU I and III clones was determined by FACS (left panel) and CCL4 concentration in supernatants from these cells was determined by ELISA (right panel) (n=3 independent transfections). f) B cells stimulated by CD40L+IL-4 were transfected to express LMP1 or EBNA2. The bar graphs show CCR1 surface expression and CCL4 release in transfected cells and in empty vector controls. Results were normalized for the percentage of transfected cells. Bar graphs show mean and standard error of the mean (n=3 transfected independent B cell samples). g) Differential CCR1 expression at the surface of B cells infected with a LMP1^{null} virus or wild type controls (left panel) and CCL4 release in the supernatants of these cells (right panel) (n=3 independent B cell samples).

Suppl. Fig. 3

Suppl. Fig. 3 CCR1 expression and function in EBV-infected B cells. ... d) EBV-negative Burkitt cell lymphoma BL41 cells were transfected with expression plasmids expressing LMP1 or EBNA2 or with a vector control. The percentage of cells expressing CCR1 was determined by FACS. CCL4 concentrations in supernatants of these cells were determined by ELISA. Results are given in pg/ml. Bar graphs show mean CCR1 or CCL4 expression with standard deviation (n=3 independent transfections). Statistical significance was determined using paired in a) and c) (motility) and unpaired t-tests in c) (velocity) and one-way analysis of variance in d).

3) While much of the authors' data on CCL4-induced changes in cellular responses are convincing, they can be inferred from the well-known functions of chemokines. In this sense, the present data are mainly confirmative.

Answer: As previously mentioned, our paper describes cellular processes (paracrine directed migration and spontaneous diapedesis) induced by the virus. Even this virus-cell interaction uses well-characterized molecules, this does not mean that it is not novel. Moreover, the observation that the virus takes advantage of the simultaneous expression of a chemokine and its receptor to induce migration is not trivial. It has been known for a long time that EBV-infected B cells produce various chemokines. However, nobody before put the available data together to reveal an important property of EBV-infected B cells that can explain important pathogenic features of the infection. Finally, we are not aware of any paper describing the highly efficient diapedesis induced by the virus independently of any exterior chemotactic stimulus, a process that is known to be central in multiple sclerosis. As mentioned above, the aim of paper is not to study chemotaxis or diapedesis per se, but to explain how EBV-infected cells use them to disseminate and how this can be used to eliminate infected B cells.

Other issues:

1. Lines 65-67: Is directional migration of B blasts induced by CXCL12 or is the differentiation state changed when primary B cells have been exposed to CX40L, IL-4, and CXCL12? To understand the roles of CXCL12 in B cell blast motility, experiments with AMD3100 should be added.

Answer: We have performed the requested experiment, exposure of CD40L+IL-4+CXCL12 stimulated B cells to the CXCR4 inhibitor AMD3100 blocked movement, confirming that CXCL12-induced B cell migration (see page 5 lines 104-107 and Suppl. Fig. 1f).

page 5 lines 104-107 and Suppl. Fig. 1f:

Similarly, CXCL12's effect on B blasts' migration largely disappeared after exposure to a CXCR4 inhibitor (AMD3100), confirming that their directional movement resulted from a chemokinetic stimulus (Suppl. Fig. 1f).

Suppl. Fig. 1

Suppl. Fig. 1 CCL3, CCL4, CCL5 attract EBV-infected B cells. ... f) The CXCR5 inhibitor AMD3100 suppresses migration of blasts stimulated with CD40L, IL-4 and CXCL12. The paths of B blasts seeded at high density in a collagen matrix in the presence or absence of AMD3100 (1 μM) were recorded by life cell imaging over 15 minutes and given as 2D tracks in their native conformation (top) or centered at the origin (bottom). The percentage of motile cells is indicated in the bar graph (n=50, one representative example from 5 experiments with independent B cell samples). Velocity and directionality was determined for motile cells. Bar graphs give means with standard deviations. Statistical analysis was done using one way analysis of variance in a) and paired t-tests in b), e) (percentage of motile cells) and f) (percentage of motile cells) and unpaired t-test in e) and f) for velocity and directionality.

2. Lines 67-68: In this experiment, do the authors observe chemotaxis or chemokinesis?

Answer: We observed chemokinesis as the cell paths generated by infected B cells are piecewise linear, but do not converge to a particular source (see page 4-5 line 87-92).

page 4-5 line 87-92:

However, because the paths generated by EBV-infected B cells paths do not converge, these cells are subjected to chemokinesis rather than chemotaxis. This suggests that the concentration of chemokines in the extracellular milieu is homogeneous, rather than concentrated in a discrete region of the culture. Thus, the chemotactic signals in a culture of

EBV-infected B cells probably originate from many members of the cell culture, if not all of them.

3. Lines 88-90: Supplementary Figure 1 does not have data for IL-10.

The authors need to validate the expression levels of human and viral IL-10 in EBV-infected B cells.

Answer: We thank the reviewer for having raised this point. We have used the EBV-encoded IL-10 knockout to distinguish and quantify cellular IL-10 and EBV-encoded IL-10. Both cytokine variants can be detected using the same Elisa. We found that EBV-infected cells release more cellular IL-10 (80% of total) than EBV-encoded IL-10 (approximately 20% of total) (see Suppl. Fig. 1a, page 5, line 93-98).

page 5, line 93-97:

ELISA-based assays confirmed that all these cytokines are secreted in the supernatant of EBV-infected B cells, with CCL3 and CCL4 being secreted at concentrations more than hundred times higher than in supernatants from primary B cell controls (Suppl. Fig. 1a).

Suppl. Fig. 1

Suppl. Fig. 1 CCL3, CCL4, CCL5 attract EBV-infected B cells. a) Concentrations of CCL3, CCL4, CCL5 and IL-10 in supernatants of resting primary B cells, B cells stimulated with CD40L+IL-4 and EBV-transformed B cells were determined by ELISA. The latter included cells transformed by wild type EBV or by an EBV-encoded IL-10 knockout (vIL-10^{null}). Purified EBV-encoded IL-10 served as a positive control. Bar graphs show mean and standard error of the mean (n=3 independent B cell samples). ...

4. Line 172: The term “viral IL-10” has not appeared in any of the previous texts, so some explanation should be added.

Answer: Thank you for spotting this. We have replaced ‘viral IL-10’ by ‘EBV-encoded IL-10’ throughout the paper (see e.g. page 5 line 93-94)

Page 5 line 93-94:

... We identified human IL-10 (hIL-10), EBV-encoded IL-10 homolog (also referred to as viral IL-10 or vIL-10),...

5. Lines 626-629: For multiple comparisons, the method of pos-hoc tests should be described as well as ANOVA. In addition, the range of P-values corresponding to the number of stars should be shown.

Answer: Fixed (see page 31, lines 733-736).

page 31, lines 733-736:

The error bars represent the standard deviation of the data sets. Statistical significance was determined using the student's t-test or ANOVA analyses combined with Dunnett's multiple comparisons test. P values are displayed by asterisks with * = $p < 0.05$, ** = $p < 0.01$, *** = $p < 0.001$ and **** = $p < 0.0001$.

6. Half-width spaces between numbers and units are missing (for example, lines 110, 327, 834, 947, 1021, 1022, Fig. 3c and d, Suppl. Fig. 2b-e, Fig. 3c).

Answer: We have corrected the paper accordingly (all over the manuscript).

Reviewer #2 (Remarks to the Author):

Epstein-Barr virus (EBV) infection of human B cells can cause human lymphomas and is also associated with the development of autoimmune diseases, particularly multiple sclerosis (MS) and lupus. The mechanisms(s) by which EBV infection of B cells might increase the development of MS are not fully understood, but may involve both increased EBV-directed antibodies that cross-react with CNS antigens, as well as increased migration of EBV-infected B cells across the blood-brain barrier. In this paper, the authors present a wealth of data suggesting that EBV-infected B cells display features of homing cells, including an increased ability to migrate through endothelial cells. They show that increased migration is mediated by increased expression of both CCL4 and CCR1 in EBV infected B cells, which results in activation of the FAK2 kinase. Furthermore, they show that a drug which inhibits the FAK1 and FAK2 kinases decreases migration of EBV-infected B cells into the CNS in a mouse model and also inhibits EBV-induced lymphomas in this model. Finally, they show that increased expression of EBV-encoded IL10 also helps to recruit uninfected primary B cells towards EBV-infected B cells. Overall, the data are of high quality and the findings are significant and novel, suggesting that FAK kinase inhibitors might be useful for treating EBV-associated diseases, including MS.

However, the paper would be improved by including the following:

1. It is not clear how many donors were used to generate the EBV-infected B cells used in the study. Did they all come from a single donor? If so, the authors need to show that similar results are obtained in at least two different EBV-infected cell lines, obtained from different donors. They should also state how long after EBV infection the studies were performed. Is the increased

migration phenotype only seen in early passage EBV-infected cell lines or is this a persistent phenotype?

Answer:

The reviewer raises an important point.

a) In all experiments we used at least three lymphoblastoid cell lines generated from independent healthy individuals. For crucial experiments such as the description of migration and diapedesis, the results were confirmed with between 5 and 7 different cell lymphoblastoid lines generated from independent healthy individuals. These numbers are now given for each experiment in the figure legends.

b) The studied cell lines have been tested at different time points after establishment. We described in the first version of the paper how the lamellipodial appendage develops early after infection and how migration progressively develops. We now include observations made until 120 days post-infection that confirm that the polarization, lamellipodia development and migration phenotype are persistent (See page 8 lines 169-171, Suppl. Fig. 4f).

page 8 lines 169-171, Suppl. Fig. 4f:

Interestingly, polarization and lamellipodia developed slowly post-infection and reached their full development only 2 weeks after infection, a time at which cells start to migrate and proliferate efficiently (Suppl. Fig. 4f). After that time point, these morphological features and the ability to migrate persisted unchanged, even after several months.

Suppl. Fig. 4

Suppl. Fig. 4 EBV-infected B cells are polarized (I). ... f) The development of the lamellipodial appendage in EBV-transformed B cells and their migrating abilities were monitored for 120 days after EBV infection. Both the ratio between the size of the lamellipodia and the size of the main cell body (left panel, n=25 one representative example from 3 experiments with independent B cell samples) and the percentage of migrating cells (right panel, n=4 independent B cell samples) are indicated. Bar graphs give the mean and standard deviation. Statistical significance was determined using one-way analysis of variance...

2. The authors should make some attempt to identify the EBV protein(s) that mediate increased CCL4 and CCR1 expression in B cells. Does this only occur in B cells with "type III" latency or

does it also occur in B cells with more stringent forms of viral latency (which are much more common in the human host). This should be possible since another group reported that CCR1 expression is increased in EBV-positive Burkitt lymphomas that have drifted to "type III" latency in vitro. At the very least they could over-express the EBV EBNA2 or LMP1 proteins in an EBV-negative or type I EBV+ Burkitt line to determine if EBNA2 or LMP1 expression mediates the phenotype they are describing.

Answer: We have used multiple experimental systems to answer this question. We first monitored CCL4 and CCR1 expression in Mutu I (latency I) and Mutu III (latency III) Burkitt's lymphoma clones. This assay showed that cells in latency III, but not those in latency I, strongly express both CCL4 and CCR1 (pages 6-7 lines 127-147 and Fig. 2 e-g and Suppl. Fig. 3d). We then transfected EBNA2 and LMP1, both in EBV-negative BL41 Burkitt's lymphoma cells and in primary B cells stimulated with CD40 ligand and IL-4. These assays revealed that both EBNA2 and LMP1 are potent CCL4 inducers. However, only LMP1, and not EBNA2, could induce CCR1 surface expression in either experimental system. To confirm LMP1's role in CCR1 and CCL4 expression, we infected primary B cells with a LMP1^{null} mutant (M81/ΔLMP1). Relative to B cells infected with wild type viruses, B cells infected with M81/ΔLMP1 released CCL4 and expressed CCR1 at 5- and 10-times lower levels than their wildtype counterparts, respectively. Altogether, we found that cells in latency III and cells that express LMP1, but not cells in latency I express both CCR1 and CCL4 (see Fig. 2 d to g).

pages 6-7 lines 127-147:

EBNA2 and LMP1 control CCR1 and CCL4 expression. Latently infected B cells proliferate under expression of the latent genes, among which the Epstein-Barr nuclear antigen 2, a transactivator that activates the Notch pathway and the latent membrane protein 1, a permanently active viral homolog of CD40, play a crucial role. To establish whether these viral genes control CCL4 and CCR1 expression in infected B cells, we first monitored their expression in an early and a late passage of the MUTU Burkitt's lymphoma cell line. While an early passage of this line (MUTU I) has a restricted latent protein expression pattern largely limited to EBNA1 (latency I), late passage cells (MUTU III) express all latent genes, including EBNA2 and LMP1 (latency III)¹⁷. CCR1 surface expression and CCL4 release in MUTU III was respectively 8 (CCR1) and 600 (CCL4) times higher than in MUTU I, suggesting that expression of these proteins is associated with latency III (Fig. 2e). To determine which latent gene is responsible for these effects, we transfected EBNA2 or LMP1 in CD40L+IL4 activated B blasts. These assays revealed that LMP1 is a potent inducer of CCL4 and CCR1 expression (Fig. 2f). Interestingly, EBNA2 also activated CCL4 release, but was unable to induce CCR1 expression (Fig. 2f). Similar results were obtained after transfection of the cell line BL41 with LMP1 or EBNA2 (Suppl. Fig. 3d). To confirm LMP1's role in this process, we infected primary B cells with a LMP1^{null} mutant (M81/ΔLMP1). Relative to B cells infected with wild type viruses, B cells infected with M81/ΔLMP1 released CCL4 and expressed CCR1 at 10- and 5-times lower levels, respectively (Fig. 2g). This approach was unfortunately not possible for EBNA2 as the M81/ΔEBNA2 mutant fails to initiate transformation. Altogether, we conclude that LMP1 and EBNA2 collaborate to initiate the chemokine loop that leads to migration.

Fig. 2

Fig. 2 Migration of EBV-infected B cells is driven by CCL4 and CCR1. ... e-g) EBV latent genes and CCL4-CCR1 expression. e) CCR1 surface expression on Burkitt cell line MUTU I and III clones was determined by FACS (left panel) and CCL4 concentration in supernatants from these cells was determined by ELISA (right panel) (n=3 independent transfections). f) B cells stimulated with CD40L+IL-4 were transfected to express LMP1 or EBNA2. The bar graphs show CCR1 surface expression and CCL4 release in transfected cells and in empty vector controls. Results were normalized for the percentage of transfected cells. Bar graphs show mean and standard error of the mean (n=3 transfected independent B cell samples). g) Differential CCR1 expression at the surface of B cells infected with a LMP1^{null} virus or wild type controls (left panel) and CCL4 release in the supernatants of these cells (right panel) (n=3 independent B cell samples).

Suppl. Fig. 3

Suppl. Fig. 3 CCR1 expression and function in EBV-infected B cells. ... d) EBV-negative Burkitt cell lymphoma BL41 cells were transfected with expression plasmids expressing LMP1 or EBNA2 or with a vector control. The percentage of cells expressing CCR1 was determined by FACS. CCL4 concentrations in supernatants of these cells were determined by ELISA. Results are given in pg/ml. Bar graphs show mean CCR1 or CCL4 expression with standard deviation (n=3 independent transfections). Statistical significance was determined using paired in a) and c) (motility) and unpaired t-tests in c) (velocity) and one-way analysis of variance in d). ...

Reviewer #3 (Remarks to the Author):

In this manuscript, Delecluse et al describe a process in which Epstein-Barr virus infected B cells upregulate chemokines and their receptors to support direct migration, recruitment of uninfected B cells, and diapedesis along with disruption of endothelial cell barrier function. Overall, this is a tour de force with many experiments characterizing the molecular details of this newly described molecular circuitry enabling a robust migratory phenotype in human B cells.

The initial findings are that EBV infection promotes a directed B-cell migration phenotype relative to CD40L/IL4 induced B blasts and uninfected B cells as evidenced by live cell microscopy and image analysis and similar in velocity and direction to B blasts treated with CXCL12. This activity is associated with upregulation of both the chemokines CCL3, 4, and 5 (MIP1a, b, and RANTES) and their cognate receptors CCR1. The authors demonstrated that EBV-infected B cells are recruited to CCL4 and that the depletion of CCL4 by antibody or gene knockout significantly reduces motility. Likewise, inhibition of CCR1 with BX471 and KO also suppresses motility. Evidence of polarized cell movement is captured by SEM and IF for cdc42 and phospho-PKCzeta as well as increased phosphorylation of FAK2. The EBV-infected cell motility requires CDC42 and ROCK activity as well as FAK2 as evidenced by inhibition with defactinib and combination experiments show potential evidence of synergy between CCR1 antagonism and FAK2 inhibition. Another fascinating finding is that EBV-infected cells recruit uninfected B cells by transwell assay. This activity is apparently driven by viral IL-10 as a knockout decreases and providing exogenous vIL-10 increases uninfected B cell movement across the transwell. Single cell sequencing characterizes the motile population as upregulated CD52, which is validated by flow cytometry. Further characterization indicates an increase in CD11c in this population.

The next set of experiments include demonstrating that EBV-infected B cells can undergo diapedesis through various endothelial cell barriers. This is very intriguing and supported by several experiments comparing to resting B cells, B blasts and CXCL12 directed motile B blasts, none of which act like EBV-infected B cells in terms of crossing the endothelial barrier or disrupting the trans-endothelial electrical resistance. These attributes and endothelial cell adhesion all rely on CCR1 and FAK2. Finally, EBV-infected B cell invasion of the spleens of NSG mice were dependent on FAK2.

Overall, there is a significant amount of data supporting the directed motility driven by CCR1/CCL4 and FAK2 dependent actin polymerization processes. Further, diapedesis by EBV-infected B cells and disruption of the TEER of endothelial cell barriers support a model whereby EBV-infected cells have the capacity to potentially breach the BBB in MS.

While most of the experiments were very well controlled and easily interpretable, there were a few that could have been either left out of the study (eg the IL10 recruitment of uninfected B cells) or bolstered with additional data (eg scRNAseq).

Minor experimental comments:

1. The scRNAseq of uninfected B cells provides an initial glimpse of what changes are driven by exposure to the EBV-infected cell supernatants and presumably a vIL-10 mediated process, but this is limited to CD52 RNA and validation by flow along with CD11c flow. It would be interesting to see what the overall changes are between motile and non-motile resting B cells.

Answer: We now show the UMAP representation of the unsupervised scRNA-seq analysis labeled by experimental condition in Fig. 4 c and d. There was no difference between the move and stay populations in terms of B cell differentiation. However, a Gene Ontology analysis showed that the move population displayed an increased transcription of genes implicated in the oxidative phosphorylation and reduced entry in mitosis (Fig. 4e). Furthermore, there was an increase in pathways characteristic of a response to interferon beta and gamma. The latter result was expected as EBV-infected cells produce these molecules. (Please see Fig. 4 c to e, See also page 10 lines 221-231).

page 10 lines 221-231:

UMAP visualization based on clustering analysis identified 3 clusters within the primary B cell population. These were defined on the basis of a NF- κ B activation signature, of a BCR response with SYK expression or of a combined IL-4R and IL-7R expression, but we did not find any evidence that this clustering differed between the move and stay population (Fig. 4c and 4d). However, gene set enrichment analysis (GSEA) revealed that moving primary B cells strongly express interferon alpha and gamma response genes, that probably results from the contact with infected B cells that produce interferons (Fig. 4e)^{27,28}, as well as an increased oxidative phosphorylation and a decreased entry into mitosis. ScRNA analysis also identified CD52 and CD53 expression as enhanced in the majority of migrating B cells, relative to their immobile counterparts (Fig. 4f, Suppl. Table 2, extended data). ...

Fig. 4

Fig. 4 EBV-infected B cells attract CD52^{high} CD11c⁺ primary B cells. ... c) Uniform Manifold and Projection (UMAP) plot of scRNA-seq analysis performed on all primary B cells and colored by annotation. d) UMAP visualization of scRNA-seq data from mobile (move) and immobile (stay) B cell populations. e) Top five enriched (orange) and depleted (blue) Gene Ontology terms in the mobile B cell population, relative to the immobile B cells. ...

2. In Figure 2c (and other similar expmts) are these B cell clusters moving or single cells?

Answer: These are migrating single cells, we have clarified this point in the revision (Please see page 4 line 72-74).

page 4 line 72-74:

The recorded trajectories of these single cells displayed an intermediate degree of directionality for both B cell subtypes (60%), supporting this hypothesis (Fig. 1d, see extended data for a detailed explanation). ...

3. Need isobologram to argue for synergy of defactinib with BX471

Answer: We have added a dose-response curve and isobolograms (See page 9, line 195-202, Fig. 3e)

page 9, line 195-202:

We confirmed synergy between BX471 and defactinib first by drawing a dose-response curve for these drugs (Fig. 3e). This analysis showed a very steep dose-response slope for the latter drug that limited the range of concentrations that could be used to draw isobolograms. Nevertheless, isobolograms for E10 (10% of complete growth reduction) and E30 (30% of complete growth reduction) showed clear synergistic effects (Fig. 3e). At higher doses, both drugs were individually able to reduce cell growth. Under these conditions, as expected, synergistic effects could not be observed. Altogether, these data confirm that CCR1 and FAK2 are located in the same pathway and are essential for LCL polarization and cell migration.

Fig. 3e

Fig. 3 Migration of EBV-infected B cells is dependent on FAK2. ... e) dose-response curve for defactinib and BX471 (n=3), together with isobolograms (ED₁₀ = 10% of maximal effect and ED₃₀ = 30% of maximal effect). The linear curve shows additivity of the drug effects, the curve below it is indicative of synergistic effects. Statistical significance was determined using paired t-tests in a) (growth curve at day 14), b) and c) and one-way analysis of variance at day 7 in d).

4. How many uninfected B cells are moving in the transwell assays? These are normalized and makes it difficult to interpret

Answer: On average approximately 2.5x10³ B cells spontaneously move to the bottom well in transwell assays, (see legend to Fig. 4a). However, there was some variation between the individual primary B cell samples we tested that required normalization as shown in this figure.

Legend to Fig. 4a:

Fig. 4 EBV-infected B cells attract CD52^{high} CD11c⁺ primary B cells. a) Transwell migration assay with EBV-infected B cells in the bottom well and labeled primary resting B cells in the top well. The graph shows the mean number of labeled primary B cells that reached the bottom chamber after 24 hours, relative to spontaneous B cell migration in the absence of infected B cells that served as a negative control (n=5 independent B cell samples, each experiment in

duplicate). On average approximately 2.5×10^3 primary B cells spontaneously migrated to the bottom chamber.

Minor text and stylistic comments:

1. In Figure 1, it would be good to change the color of the CD40L/IL4/CXCL12 from that of CD40L/IL4 as it's difficult to distinguish the two.

Answer: We have changed the colors as suggested, see Figure 1.

Fig. 1

2. It would be easier to follow if the time courses were described initially for the B blast and EBV infections to understand what the comparisons are.

Answer: Fixed, see Fig. 1a, b and c.

Fig. 1

3. Which band is the phospho-FAK2 in Fig 3c? Presumably the one with the asterisk, but usually an arrow would mark the actual band and an asterisk perhaps the background band above it that doesn't change.

Answer: We have added an arrow as suggested and kept the asterisk for the non-specific signal, see Fig. 3c.

Fig. 3

4. The data presented in fig 5c and d should be quantified

Answer: We have quantified these data, see Fig. 5c and d.

Fig 5.

Fig. 5 EBV-infected B cells undergo diapedesis. ... c) HBMEC and HUVEC endothelial cells were cocultured with EBV-infected B cells for 30 minutes. The effect of coculture on ZO-1 expression was evaluated 24 hours later by immunofluorescence. Its localization was determined by intensity profiling of gray values, membrane peaks being marked with arrows (middle panel). Bar graphs give the percentage of cells that have a membrane peak (n=5 independent B cell samples). d) Same as in c), but cells were analyzed for ICAM-1 expression. The bar graphs give the raw integrated density of fluorescence (Raw/Int/Den) in the different samples (n=5 independent B cell samples). ...

5. Line 96 'CCL4'

Answer: Thank you for spotting the mistake, it has now been corrected.

6. Page 5, line 59 'very high' relative to what?

Answer: Relative to uninfected B cells. This has been clarified, see page 5 line 93-98.

page 5 line 93-98:

We identified human IL-10 (hIL-10), EBV-encoded IL-10 homolog (also referred to as viral IL-10 or vIL-10), CCL3, CCL4 and CCL5 as potential candidates¹⁵. ELISA-based assays confirmed that all these cytokines are secreted in the supernatant of EBV-infected B cells, with CCL3 and CCL4 being secreted at concentrations more than hundred times higher than in supernatants from primary B cell controls (Suppl. Fig. 1a). ...

Rebuttal letter

Reviewer #1 (Remarks to the Author):

I understand the novelty of this study in the revised version. Although the data presented are primarily confirmatory regarding the well-known functions of chemokine and chemokine receptors, the experiments were accurately performed and are reliable. The potential molecular mechanism on induction of chemokines by EBV-infected B cells has now been included in the text and Figures.

minor issues:

1. Supplemental Fig. 1 (f) , line 25: “CXCR5” must be “CXCR4”.

Answer: We fixed this issue.

2. Supplemental Fig. 1 (f) , line 28: “life cell imaging” must be “live cell imaging”.

Answer: We fixed this issue.

Reviewer #2 (Remarks to the Author):

This revised paper has adequately addressed the points made by the three previous reviewers. The manuscript (which was already highly significant and original) is now even further improved.

Reviewer #3 (Remarks to the Author):

The authors have satisfactorily responded to my concerns. The manuscript is improved and in excellent shape for publication. This study should have a large and sustained impact on the EBV field.